# Sulfur-containing class of broad-spectrum antivirals improves influenza virus vaccine development

David W. Buchholz [1,16], Armando Pacheco[1,2,16], Sreetama Pal[3],
I. Abrrey Monreal[1], Shi Xu [2,4], Brian Imbiakha[1], Julie Sahler [1], Mason Jager [1],
Alex Liqi Lai[5], Erik M. Contreras[1], Shahrzad Ezzatpour[1], Brandan Cook[2],
Elshan Ralalage[6], Qian Liu[7], Yao Yu Yeo[1], Andrew Ma [1], Haewon Byun[1],
Obaed Shah[1], J. Lizbeth Reyes Zamora [1], Niraj K. Shil[8], Sara Jones-Burrage[9],
Suzanne M. Pritchard[8], Chuntao Yang[2], Yu Zhao[2], Zeinab J. Mohamed[3],
Cheyan Xu[3], Michael J. Jung[6], Gerlinde R. Van de Walle [1],
Suchetana Mukhopadhyay [9], Masako Shimamura[10], Alan G. Goodman [11],
Michele Hardy[12], Santanu Bose[8], Anthony V. Nicola[8], Jack H. Freed [5],
Avery August [1], Susan Daniel [3], Petr Chlanda [13], Jace W. Jones [14],
Ming Xian [2,4] & Hector C. Aguilar [1,15] ✉

Enveloped viruses are significant zoonotic disease threats with the potential to cause global pandemics. We identified a class of small-molecule sulfur-containing antiviral compounds (XM series) that broadly inhibit enveloped viruses. Multidisciplinary approaches revealed that XM compounds alter the viral membrane lipid chemical composition, enhance membrane order within the hydrophobic bilayer, and increase membrane phase transition temperatures. This mechanism inhibits membrane fusion and viral entry, while leaving the viral glycoproteins and genomes largely unaffected. Leveraging these unique properties, we develop a proof-of-concept whole inactivated influenza virus (IIV) vaccine using XM-01 (XM-01-IIV). In a mouse model, XM-01-IIV elicit significantly enhanced neutralizing antibody responses against hemagglutinin and neuraminidase compared to traditional paraformaldehyde-inactivated vaccines. Further, XM-01-IIV reduces morbidity and mortality following influenza challenge, achieving protection comparable to live virus vaccination. This promising class of broadly acting antivirals can be highly impactful in the development of highly potent inactivated vaccines for enveloped viruses.

The 2020 World Health Organization's (WHO) list of emerging pathogens most likely to cause pandemics and require urgent research and development consists entirely of enveloped viruses, including Ebola, SARS-CoV-2, and Nipah (NiV) viruses[1]. These viruses are responsible for millions of deaths and impose economic burdens amounting to trillions of dollars annually[2–4]. While established vaccines reduce the human and economic cost of viral outbreaks, new vaccine candidates against emerging pathogens do not always generate adequate protection from illness[5]. Therefore, the ability to rapidly generate effective vaccines for emerging or established viruses is imperative.

Whole inactivated virus (WIV) vaccines offer key advantages over live attenuated virus (LAV) vaccines, including replication incompetence, which enhances safety, and the presentation of a broader repertoire of antigens compared to subunit or genetic vaccines[6,7]. However, WIV vaccines frequently suffer from insufficient immunogenicity or, in some cases, vaccine-enhanced disease. These issues often stem from conformational changes in viral antigens introduced during inactivation processes, which typically involve chemical (e.g., formalin or β-propiolactone [BPL]) or physical (e.g., heat, UV light, irradiation) methods[7–16]. Developing methods that fully inactivate viruses while preserving the native, immunogenic conformations of glycoproteins is therefore of paramount importance.

Influenza virus (IV), which causes up to a billion infections and 350,000–650,000 deaths annually, exemplifies the challenges of WIV vaccines. Current IV vaccines achieve modest efficacies of 10–60%[17,18], largely due to limited heterologous immunity and diminished immunogenicity arising from the inactivation process[13,16,19], despite a vaccine including 3–4 IV strains. WIV vaccination shortcomings often point to the chemical methods of virus inactivation. For example, the properties that make formalin-inactivated vaccines ideal to inactivate a wide range of viruses, namely non-specific cross-linkers that react with amino acids, RNA, and/or DNA, can also cause destruction, alteration, or sub-optimal presentation of antigens to the immune system[11,15,16]. Indeed, in early clinical trials, formalin-based WIV vaccination of respiratory syncytial virus (RSV) resulted in enhancement of disease due to impaired glycoprotein structure[20,21]. Maintaining native glycoprotein structure is thus key to WIV vaccine development.

Over the past decade, several potential broad-spectrum antivirals were identified which act specifically on the viral membrane, often via light activation, including LJ001, JL122, and the hydrogen sulfide ($H_2S$)-producing compound GYY4137. These compounds display inhibitory properties against multiple enveloped viruses, including respiratory syncytial virus (RSV), human metapneumovirus (hMPV), NiV, and others[22–24]. Targeting the viral membrane is an attractive option in preventing viral entry/infection; however, the abundance of host cellular membranes may limit the use of such compounds as therapeutics.

In this study, we identified a class of small-molecule antiviral compounds (XM series) that target the viral membrane while maintaining the integrity of viral surface glycoproteins, characteristics ideal for vaccine development. A multi-disciplinary characterization of the biological activity of XM compounds indicated that these light-independent persulfide-radical producing agents embed into viral membranes, alter their biophysical and chemical properties, and inhibit viral-cell membrane fusion, viral entry, and productive infection. To test whether these characteristics of XM compounds made them suitable for WIV vaccine development, we optimized the inactivation of an influenza virus strain and corroborated that XM treatment completely inactivated the virions without affecting viral glycoprotein functions. Further, mice vaccinated with inactivated influenza virus (IIV) via XM-01 developed more highly neutralizing antibody responses to both hemagglutinin (HA) and neuraminidase (NA) glycoproteins, and upon viral challenge yielded reduced morbidity and mortality compared to other IIV approaches. Finally, we demonstrated that inflammation and immune responses in XM-WIV vaccinated mice at 5 days post challenge were comparable or improved relative to other vaccine groups. Thus, lipid-membrane targeting XM class of compounds could lead to high potential for the development of enveloped virus whole inactivated vaccines.

## Results

### XM compounds inhibit enveloped virus infections

We screened for antiviral activity a library of sulfur-containing compounds in comparison to known $H_2S$ donors, NaHS and GYY4137[25,26], and to a light-dependent inhibitor of membrane fusion and viral entry, LJ001 (Figs. 1A, and S1)[23,27]. These compounds were first evaluated using a pseudotyped Nipah virus (NiV)/vesicular stomatitis virus (VSV)-based *Renilla* luciferase reporter infection system (pNiV)[28–30], in which the NiV attachment (G) and fusion (F) glycoproteins replace the VSV surface glycoprotein G, and permit a single round of viral entry/infection. This reporter system permits high throughput quantitative testing of antivirals. Pretreatment of pNiV with compounds at 10 μM concentration for 30 min prior to infection of Vero cells showed a significant decrease in virion infectivity for five XM compounds (Fig. 1A). Interestingly, all five active compounds (XM-01, -02, -03, -06, and -12) contain acyl disulfide moieties, potentially enabling persulfide radical formation (Fig. S1).

We then evaluated the cytotoxicity levels of these five inhibitory XM compounds in Vero cells using a CCK-8 cytotoxicity kit at 1, 3, and 10 μM for 30 min–24 h (Fig. 1B). Based on these results, we proceeded with XM-01 for further characterization. XM-01 cytotoxicity was then tested from 1 μM to 1 mM in Vero cells (Fig. 1C) and 1–100 μM in MDCK cells (Fig. S2) for 24 h, showing that XM-01 did not significantly affect cell viability at concentrations <300 μM. Further, XM-01 inhibited the enveloped pNiV virus, herpes simplex virus 1 (HSV-1), respiratory syncytial virus (RSV), vesicular stomatitis virus (VSV), or influenza A/California/04/2009 (Ca09) virus (Fig. 1D) with $IC_{50}$s between 1 and 40 μM, yielding selectivity indexes ($CC_{50}/IC_{50}$) between 1:25 and 1:1000 depending on the virus. This large variation between viruses may be due to the differential density of viral-surface glycoproteins or lipid composition of viral membranes. In contrast, entry of the non-enveloped rotavirus (Fig. 1D) was not inhibited. Further, XM-01 was able to function in the absence of light, which is a required trait among other compounds of a similar class (Fig. S3). Collectively, our results indicate that XM-01 broadly inhibits enveloped viruses regardless of viral genome type or viral family, suggesting that viral membranes may be the primary target of XM-01 action.

### XM-01 perturbs viral membranes without affecting viral glycoproteins or RNA

To determine if viral infection is inhibited by XM-01 at viral entry or post-viral entry steps, we incubated Vero cells with XM-01 at different time points post-viral infection. Cells were infected with pNiV for 2, 4, 6, 8, 18, or 24 h, after which unbound pNiV was washed away from the cells, and 100 μM XM-01 in media was added for the remainder of 24 h at 37 °C. pNiV infection was then measured via luciferase activity (Fig. 2A). No inhibition by XM-01 was observed compared to the DMSO vehicle control at any time point, indicating that once the virus has entered cells, XM-01 does not exert inhibitory activity, consistent with XM-01 specifically inhibiting the viral entry step. We next incubated either Vero cells or pNiV for 30 min with XM-01 at various concentrations prior to viral infection (Fig. 2B). Viral infection assessed 24 h post-infection (hpi) was decreased only when pNiV virions, but not cells, were pre-incubated with XM-01, suggesting a direct effect of XM-01 on virions rather than on host cells.

Further, to narrow down whether viral membranes or viral glycoproteins were affected by XM-01, we tested whether XM-01 affected the conformations of NiV-F or -G. pNiV virions were incubated with 10 μM XM-01 for 30 min, followed by measurement of protein function or by binding of established conformational antibodies to NiV F or G via flow virometry, to determine potential conformational changes in F or G[29,31–33]. As a control, we used LJ001, a compound known to affect viral membranes but not viral glycoprotein conformations[23,27]. In both cases, no statistically significant changes were observed in conformational antibody binding to either glycoprotein (Fig. 2C), indicating that native glycoprotein conformations remained conserved upon XM-01 treatment. To further test for potential effects on protein function following inactivation, we used influenza virus (IV) Ca09. When IV was treated with up to 1000 μM XM-01 for 4 h, neither neuraminidase (NA) nor hemagglutinin (HA) activity was significantly impacted as compared to the untreated control (Fig. 2D, E). These were the same

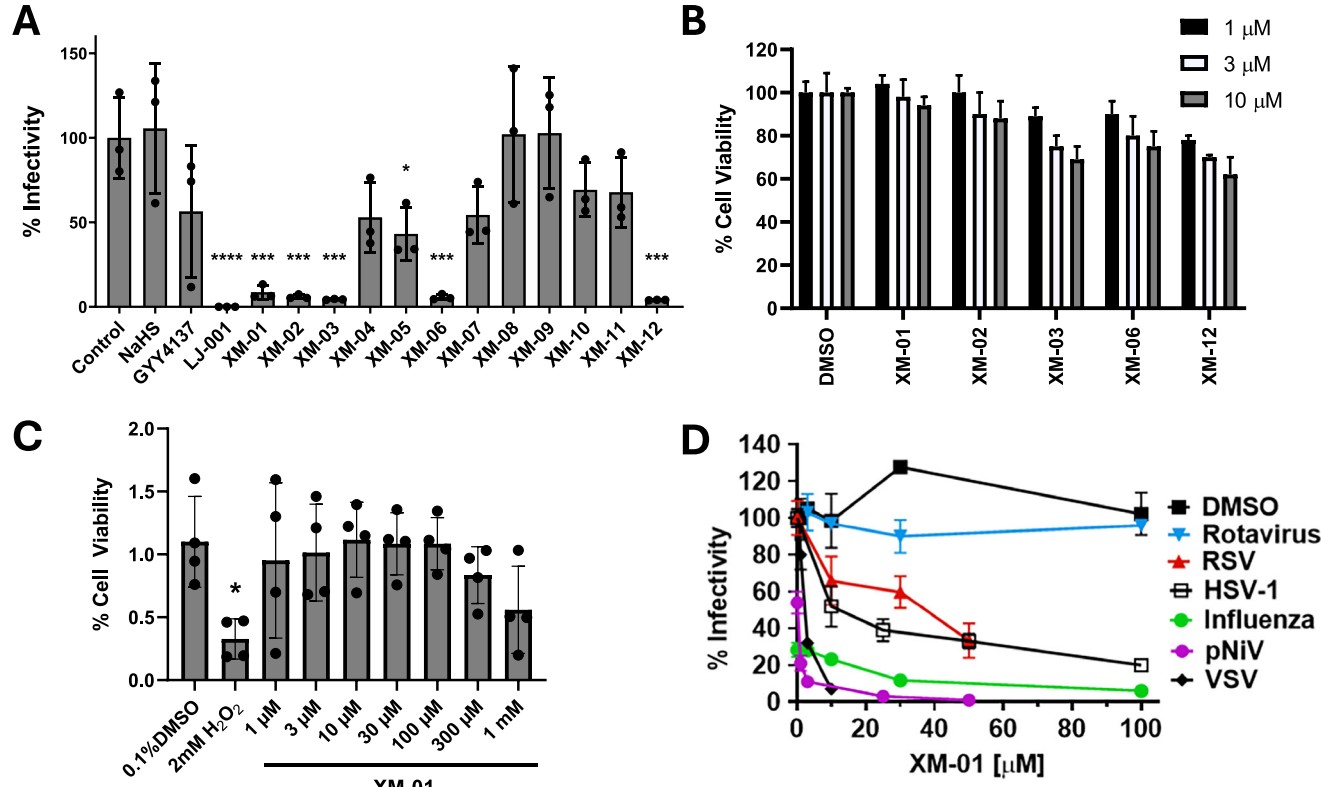

**Fig. 1 | XM-01 broadly inhibits enveloped viruses without significantly decreasing cell viability. A** Inhibition of viral infectivity using pNiV reporter virus at 10 μM concentration of compound in Vero cells. **B** Effects of five active XM antiviral compounds on cell viability, determined measuring dehydrogenase activity of live cells. **C** Vero cell viability following XM-01 treatment at 1–1 mM.

**D** Inhibition of non-enveloped (Rotavirus) and enveloped (RSV, HSV-1, Influenza, pNiV, and VSV) viruses following XM-01 viral treatment. Data are presented as mean values +/− SD. Data represent a minimum of N = 3 independent biological replicates. Statistical significance determined using t-tests or one-ANOVA and Dunnett's multiple comparison tests (ns not significant, *P < 0.05, **P < 0.01, ***P < 0.001).

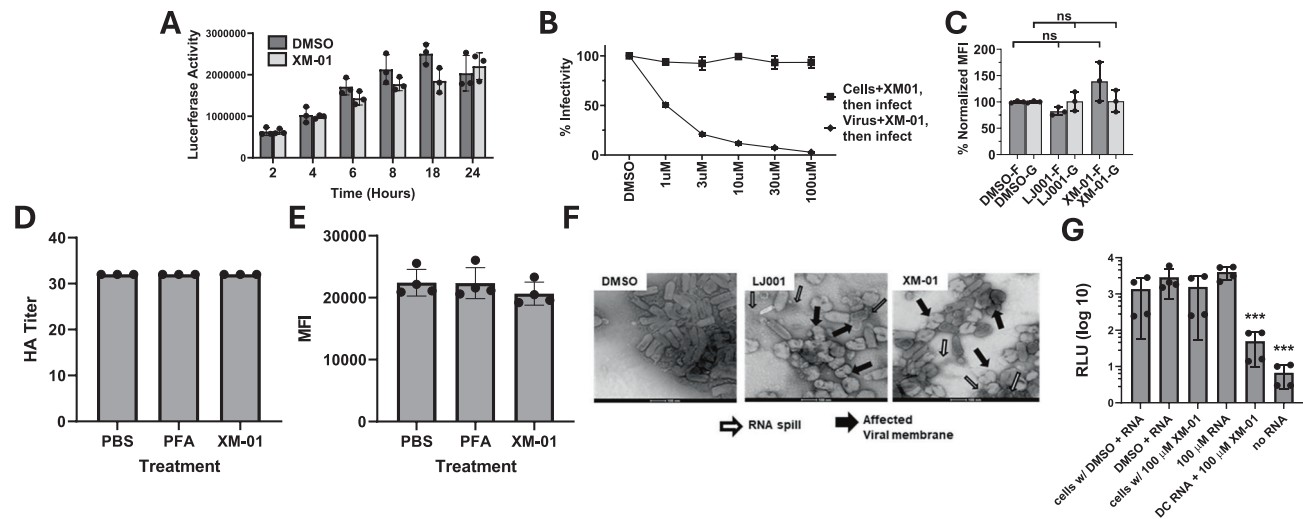

**Fig. 2 | XM-01 inhibits viral entry while preserving viral glycoprotein conformations and RNA integrity. A** Luciferase assay readout following addition of XM-01 or DMSO to cells post-infection with pNiV reporter virus. **B** Addition of XM-01 on cells or virus pre-infection shows significantly increased susceptibility of viruses to treatment over cells. **C** Preserved structure of NiV F and G glycoproteins measured via conformational antibody binding. **D** No impact of XM-01 treatment of influenza CaO9 virions on the hemagglutination activity. **E** XM-01 inhibition of

influenza virus A/California/04/09 (CaO9) strain exerts no significant impact on neuraminidase (NA) enzymatic activity. **F** Electron microscopy images of XM-01 treated virions. **G** nsp3:luciferase RNA assay shows that XM-01 treatment does not affect on RNA functional integrity. Data are presented as mean values +/− SD. Data represent a minimum of N = 3 independent biological replicates. Statistical significance determined using t-tests or one-ANOVA and Dunnett's multiple comparison tests (ns not significant, *P < 0.05, **P < 0.01, ***P < 0.001).

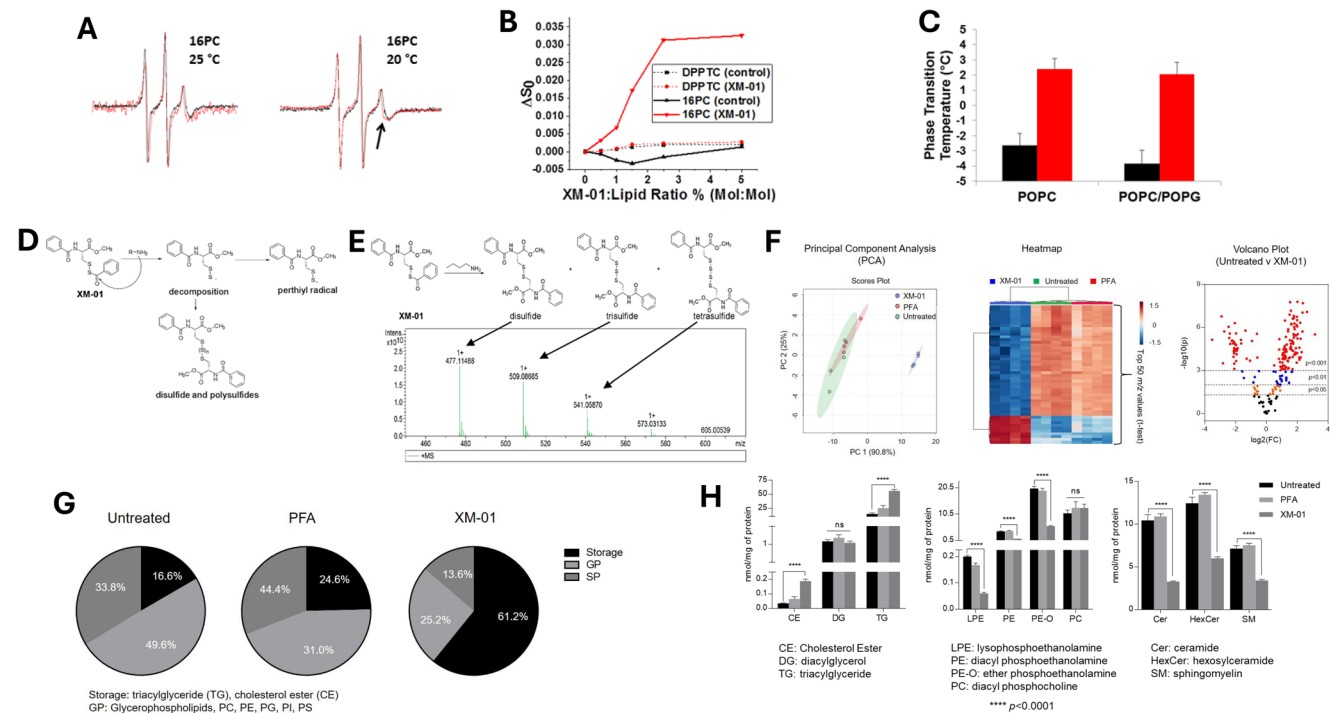

**Fig. 3 | XM-01 mechanism of action involves induction of changes in viral membranes. A** Presence of XM-01 in POPC bilayers leads to changes in electron spin resonance spectra (shown by a black arrow) of 16PC, a lipophilic spin label that resides in the membrane hydrophobic core. **B** Changes in ESR spectra quantified as an increase in membrane order ($\Delta S_0$) at various XM01:lipid ratios for DPPTC and 16PC spin-label containing liposomes. **C** Effect of XM-01 on phase transition temperature of POPC and POPC/POPG (4:1, mol/mol) membranes (P < 0.005). **D** Mechanism of generation of disulfide and polysulfide molecules, leading to perthiyl radicals, by XM-01. **E** Polysulfide and persulfide intermediate (RSSH) formation from XM-01 indicated by mass spectrometry. **F** LC-MS/MS lipid analysis displaying the top 50 most differentially expressed lipids upon XM-01 treatment compared to PFA or untreated influenza virus. N = 4 per group, data were sum normalized, log transformed, and mean-centered. Volcano plot highlighting features that had a *P < 0.05 (orange), **P < 0.01 (blue), and ***P < 0.001 (red) when comparing untreated and XM-01. The x-axis is log2(FC) (FC fold change), and the y-axis is −log10(p) (p = p-value based on t-test). **G** LC-MS/MS lipid statistical analyses showing percentages of total lipid abundance for XM-01 vs. control groups, and for **H** specific lipid class abundances for these groups. Data are presented as mean values +/− SD. Data represent a minimum of N = 3 independent biological replicates. Statistical significance determined using t-tests or one-ANOVA and Dunnett's multiple comparison tests (ns not significant, *P < 0.05, **P < 0.01, ***P < 0.001).

conditions confirmed to yield a fully-inactivated influenza virus (Fig. 2D), and used for vaccine development as further described below for Figs. 4, 5, and S6.

To visually look for gross viral structural changes, we treated pNiV with 10 μM XM-01 or LJ001 control compound for 30 min, then imaged particles via electron microscopy. Most viral particles treated with XM-01 or the LJ001 control displayed compromised membranes, altered morphologies, and frequent RNA spillage compared to the DMSO control, consistent with XM-01 and LJ001 affecting viral membranes (Figs. 2F and S4)[23,27]. To test the effect of XM-01 on viral RNA integrity, we in vitro transcribed the CHIK reporter virus, which contains luciferase in the hypervariable domain of non-structural protein 3 (nsP3). RNA was treated with different compounds, including XM-01 and decapping enzyme, and then electroporated in BHK cells. After 2 h, enough time for the CHIK reporter to initiate translation of non-structural protein and the luciferase to be produced, cells were lysed and luciferase activity measured. No effects of XM-01 were observed for XM-01 on viral RNA (Fig. 2G). We also treated IV with 1 mM XM-01 for 4 h, then measured viral binding to POPC/Chol/GM1 liposomes by cryoelectron microscopy. We observed that viral particles remain intact and there were no changes in virus binding to liposomes, indicating that XM-01 treatment maintains the structure of the particle while preserving protein function (Fig. S5). Combined, these data indicate that XM-01 modulates viral entry by affecting viral membranes and not the viral glycoproteins or RNA.

## XM-01 alters viral membrane physical properties via persulfide radical formation

To assess XM-01's impact on viral membrane integrity, we performed a sulforhodamine B (SRB) leakage assay. SRB, encapsulated at high semi-quenched concentrations within influenza virus (IV) samples treated with XM-01 and appropriate controls, exhibited very low fluorescence signals. Any compromise in membrane integrity would result in SRB leakage and increased fluorescence. However, fluorescence intensity remained unchanged over time across all samples, indicating that XM-01 does not disrupt viral membrane integrity (Fig. S6).

We next investigated XM-01's effects on viral membrane biophysical properties, hypothesizing a radical-mediated mechanism similar to compounds like LJ001 and JL122[23,27]. To investigate how XM-01 affects the physical state of viral membranes, we used electron spin resonance (ESR) spectroscopy. This technique enables the detection of changes in membrane order and fluidity by measuring the behavior of spin-labeled lipids incorporated at different positions within the bilayer. ESR analysis allows us to determine whether XM-01 increases membrane rigidity or alters membrane dynamics, both of which could impact viral fusion. Our results show that XM-01 intercalates deep into the lipid bilayer, particularly at the hydrophobic interior, as evidenced by shifts in the spectra of spin-labeled 16PC lipids. These findings indicate that XM-01 increases the order and rigidity of the viral membrane, which is likely to reduce membrane fusion capacity (Fig. 3A, B)[34–41]. The effect was temperature-dependent, with more pronounced hydrophobic interactions at 25 °C compared to 20 °C.

To quantitatively assess these changes, we calculated the lipid order parameter ($\Delta$S0), a value that reflects the degree of organization and rigidity within the membrane. Higher $\Delta$S0 values correspond to greater membrane order. We observed that XM-01 treatment led to significant increases in $\Delta$S0 for 16PC-labeled bilayers, confirming that XM-01 promotes a more ordered and less fusogenic membrane state (Fig. 3B). Additionally, XM-01 increased the phase transition temperature of POPC and POPC/POPG membranes by ~5.3 °C and ~5.8 °C, respectively, suggesting a shift to a more gel-like, ordered phase (Fig. 3C). These changes are consistent with reduced membrane fusion capacity. We further explored XM-01's mechanism by examining its role in inhibiting viral fusion using a lipid-mixing assay with octadecyl rhodamine B (R18)-labeled IV and plasma membrane vesicles (PMVs) from HEK293T cells. XM-01 treatment reduced IV fusion with PMVs to ~5%, compared to ~21% in untreated controls, confirming XM-01-mediated inhibition of viral fusogenicity (Fig. S7). To elucidate the molecular basis of XM-01-mediated virus inactivation, we examined its chemical properties. Similar to other H2S donors, XM-01's acyl groups on sulfur are likely transferred to nucleophiles, generating persulfides that can oxidize into perthiyl radicals[42]. To confirm this, we reacted XM-01 with butylamine in $CH_2Cl_2$ and recovered cysteine polysulfides, the decomposition products of persulfides (Fig. 3D, E). The formation of these products indicates the presence of a persulfide intermediate (RSSH) derived from XM-01. In cellular environments, such persulfides are known to produce radicals[43], which suggests that XM-01 and its perthiyl radical products chemically modify viral membranes via radical formation. This mechanism was further validated using lipid-mixing assays, where IV treated with XM-01 and the radical quencher α-tocopherol exhibited higher bulk fusion compared to IV treated with XM-01 alone (Fig. S8). Of note, the XM-01 concentration used with α-tocopherol (10 μM) was lower than in other assays (1 mM; Fig. S7).

To determine how much XM-01 remains associated with the viral membrane after treatment, we utilized absorption spectroscopy. This technique measures the absorbance of light at specific wavelengths corresponding to XM-01 and allows us to estimate its concentration within the viral sample. After treatment with 1 mM XM-01, we calculated that approximately 0.31 mM XM-01 in IV viral membranes (Fig. S9), reflecting the membrane-bound fraction that persists after incubation. This estimate of the absolute XM-01 concentration may be a slight overestimation due to the membrane-induced non-linear scattering artifact that manifests as a shoulder (at ~290–300 nm) in the XM-01 absorption spectrum in IV membranes. Since XM-01 intercalates within the lipid bilayers and is able to generate persulfide radicals, we hypothesized that XM-01 may cause lipid chemical changes in membranes. To test this, we performed lipidomic analyses on fully inactivated IV, using 1 mM XM-01, 0.02% paraformaldehyde (PFA), or 10% DMSO as a negative control, all treated for 4 h at RT. The extracted membrane lipids were analyzed via liquid chromatography and tandem mass spectrometry (LC-MS/MS), and data analysis showed several classes of lipids to be markedly affected by XM-01 treatment (Fig. 3F–H), including changes in ratios of key lipids identified as crucial for initial steps in membrane fusion. These phospholipid shifts are most likely due to oxidative modification of membrane lipids by XM-01's radical-generating activity during inactivation, rather than selective partitioning or metabolic effects, since all experiments were performed on purified viral particles in the absence of cellular metabolism. These data suggests that XM-01 acts directly upon membrane lipids to change their relative distribution/presence, both at the lipid class and individual lipid structure levels (e.g., acyl chain length and degree of unsaturation). Strikingly, the ratio of phosphatidylcholine (PC) to phosphatidylethanolamine (PE) lipids (PC:PE), required to be ~0.9–2:1 for efficient membrane fusion (stalk formation), changed drastically in XM-01 treated membranes (Fig. 3G, H)[44]. The PC:PE ratio in XM-01 treated virus increased to 10.97:1, as the concentration of PE fell sharply. This may be due to a significant

portion of PE species in mammalian membranes being plasmalogens, which alleviate oxidative stress via free radical scavenging[45]. Overall, these changes are consistent with a decreasing amount of intrinsic negative curvature upon XM-01 treatment, making fusion stalk formation and thus membrane fusion energetically unfavorable.

## Vaccination with XM-01-inactivated virus improves neutralizing antibody responses and protection against live IV challenge

As a proof of principle, we sought to exploit the unique properties of XM-01 described in this study, namely being a membrane fusion antiviral that leaves viral glycoproteins largely unaffected, to produce an improved inactivated influenza virus (IIV) vaccine. We optimized complete inactivation of IV with XM-01, JL-122[23], and 0.02% PFA treatment (a traditional viral inactivation process). Treating $1 \times 10^5$ PFU/mL IV with 1 mM XM-01, 0.02% PFA, or 10 μM JL122 for 4 h at RT completely inactivated IV, while DMSO vehicle treatment did not significantly inhibit virus viability, confirmed via plaque assays and 3 supernatant passages of treated virus in chicken eggs (Fig. S10). Importantly, conditions enabling complete inactivation using XM-01 did not interfere with functions of HA or NA (Fig. 2D, E).

We verified that an intramuscular (IM) injection of 1 mM XM-01 had no effect on weight loss in mice (Fig. S11). We then vaccinated groups of 10 mice (5 males + 5 females) IM with XM-01-IIV, JL-122-IIV, PFA-IIV, live IV, or phosphate buffer saline (PBS) as a mock control group, all mixed 1:1 with alum. Although live IV IM vaccination is not approved for human use, while intranasal (IN) IV challenge is lethal in mice, IM vaccination using live virus in mice does not lead to pathology/infection, and is a benchmark for ideal IV vaccines to achieve in mouse models, providing a robust and protective immune response[46]. Mice received 2 vaccinations, 2-weeks apart, and serum was collected 7–10 days post-vaccination to assess neutralizing antibodies (NAbs) using hemagglutination inhibition (HI) assays, neuraminidase inhibition (NI) assays, and plaque reduction neutralization tests (PRNT) (Fig. 4A–E)[47,48]. Impressively, the XM-01-IIV vaccinated mice generated significantly higher NAb titers against HA and NA compared to the PFA-IIV or JL-122-IIV vaccinated mice, and performed nearly as well (for NA NAbs, Fig. 4B) or as well (for HA NAbs, Fig. 4C) as the live-virus IM vaccinated mice (Fig. 4B–E). These results are particularly noteworthy, as most conventional IIV vaccines do not induce strong anti-NA NAb responses[49], and NAbs for the XM-01 treatment were significantly improved over the PFA group at the higher Ab concentrations (Fig. 4B). Notably, in female mice after two vaccinations, XM-01-IIV NAbs against both HA and NA were even superior to those generated by the live-virus vaccine (Fig. S6A), and after a third dose, the NAbs for the live IM vaccine reached the levels of the XM-01 vaccination NAbs (Fig. S12). PRNT assays confirmed that all vaccination groups neutralized the homologous IV strain (A/California/04/2009) for all vaccination types (Fig. 4D). Remarkably, in comparison to the PFA-IIV or live vaccinations, we observed neutralization to a heterologous IV strain (A/WSN/1933) for vaccination groups JL122 and XM-01 (Fig. 4E). To test preference for antibody isotype switching, if any, between vaccine strategies, flu-specific serum antibodies from primed and boosted mice were analyzed for their isotype. Also notably, compared to mock vaccinated mice, live virus and XM-01 vaccination were the only two forms of vaccination that induced a significant increase in IgA compared to mock controls, whereas PFA inactivation created significantly more IgM antibodies (Fig. 4F).

After these encouraging results, all mice were then challenged intranasally at 5 $LD_{50}$ (1000 PFU/mouse) with IV. 100% of mice vaccinated with the XM-01-IIV or live-virus vaccines survived the challenge, whereas only 80% of PFA-IIV vaccinated mice survived, and all mock vaccinated mice succumbed to the infection within 6 days post-challenge (dpc) (Fig. 4H). Weight loss is a key predictor of IV morbidity in mice[50]. Importantly, mice vaccinated with XM-01-IIV had no

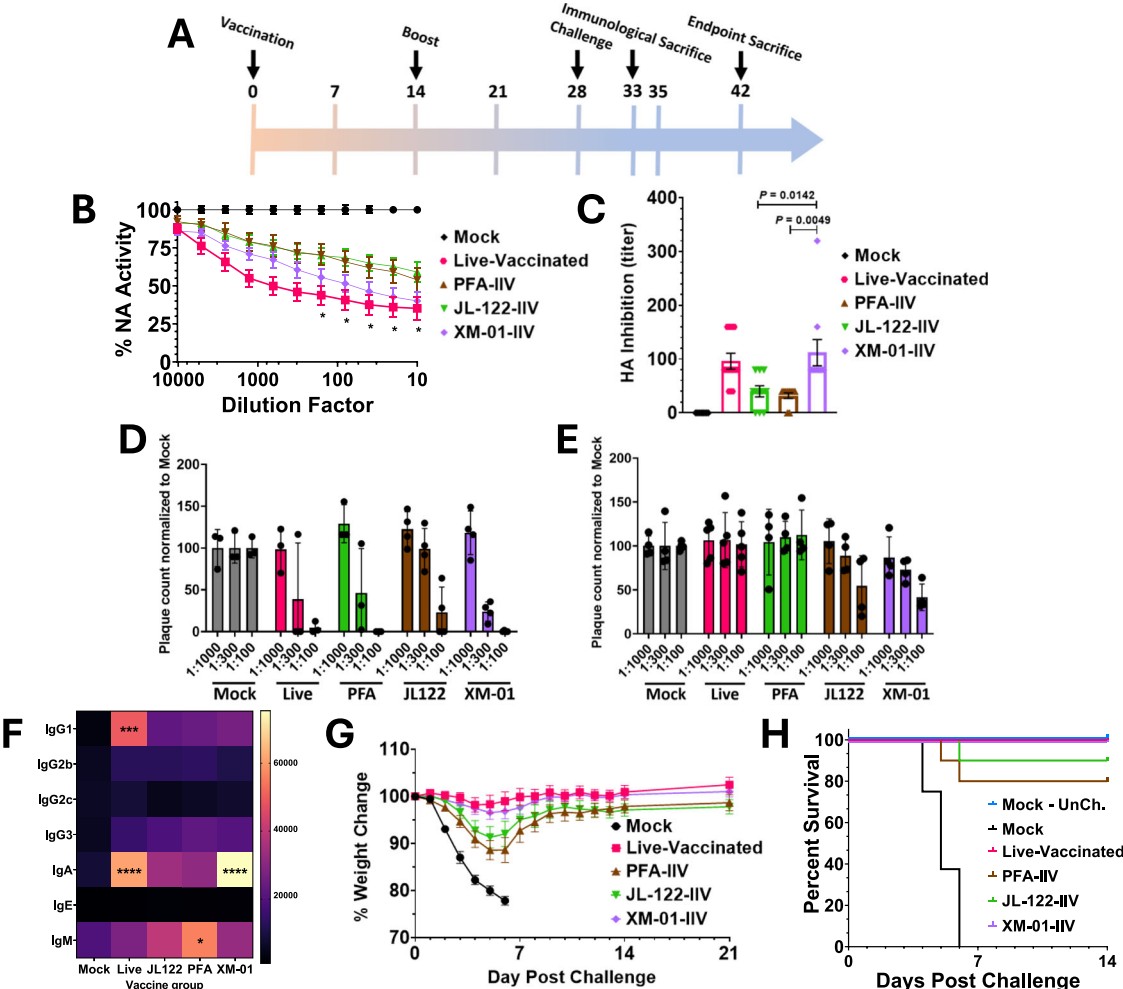

**Fig. 4 | XM-01-IIV produces an effective vaccine. A** Timeline of mouse study. **B** NA enzymatic activity of influenza virus following incubation with day-25 serum from vaccinated mice. **C** Mean of hemagglutination of red blood cells by influenza virus following incubation with day-25 serum from vaccinated mice. **D** Number of plaques from homologous IV A/California/04/2009 following incubation with 1:1000, 1:300, or 1:100 dilution of serum. **E** Number of plaques from heterologous IV A/ WSN/1933 following incubation with 1:1000, 1:300, or 1:100 dilution of serum. **F** Class of antibodies present in serum. **G** Weight loss of mice following viral challenge. **H** Survival of mice following viral challenge. N = 10 mice (5 males and 5 females) per group. Data are presented as mean values +/− SD. Significance assessed using standard t-tests (* P < 0.05, ** P < 0.01).

mortality and significantly lower morbidity than the PFA-IIV group and was comparable to the live-virus control (Fig. 4G, H).

In a complementary experiment, animals were euthanized at 5 dpc for histological analysis of various tissues, and flow cytometric analyses of immune cells (Fig. 5). Histological examination revealed peribronchiolar and perivascular inflammation with lymphocytes and granulocytes in all mice (Fig. 5A–J). Mice vaccinated with mock or JL-122-IIV vaccines developed more severe peribronchiolar and perivascular inflammation with degenerate neutrophils and cellular debris in airways (Fig. 5F, H). Mice vaccinated with PFA-IIV, XM-01-IIV, and live-virus had less inflammation and no signs of intraluminal degenerate neutrophils or debris in airways (Fig. 5G, I, J). Flow cytometric analysis of immune cells corroborated the observation of reduced lung inflammation in the PFA-IIV, XM-01-IIV, and live-vaccinated groups when compared to JL-122-IIV and saline-vaccinated groups, with fewer lung neutrophils and CD8$^+$ T cells (Fig. 5K-L). Additionally, the proportion of germinal center B cells was increased in the mediastinal lymph nodes of PFA-IIV, XM-01-IIV, and live-vaccinated groups compared to mock-vaccinated control (Fig. 5M, N), indicating a positive and specific response to vaccination.

Overall, viral inactivation using XM-01 generated a more effective IIV vaccine compared to traditional PFA treatment, by eliciting a potent relatively more protective neutralizing antibody immune response, correlating with reduced morbidity and mortality upon viral challenge.

## Discussion

We report a class of broad-spectrum antivirals with broad inhibitory properties against multiple enveloped viruses *in cellulo* and demonstrate its use as a viral inactivator for influenza vaccine development. Compounds XM-01, 02, 03, 06, and 12 possess antiviral activity and display low levels of cytotoxicity. A series of mechanistic analyses demonstrated that XM-01 affects the virus membrane chemically and physically, inhibiting viral entry, without significantly affecting the viral glycoproteins or RNA integrity. We therefore explored the potential of XM-01 for improved WIV vaccine development, as one of the major reasons for the poor performance of many WIV vaccines is the damage of the viral glycoprotein antigens during the inactivation process[8–10,12,15,16,19,51–53]. Using a mouse IAV model, we demonstrated that XM-01 treatment improved the in vivo generation of NA and HA neutralizing antibodies, in turn lessening both morbidity and mortality upon viral infection challenge.

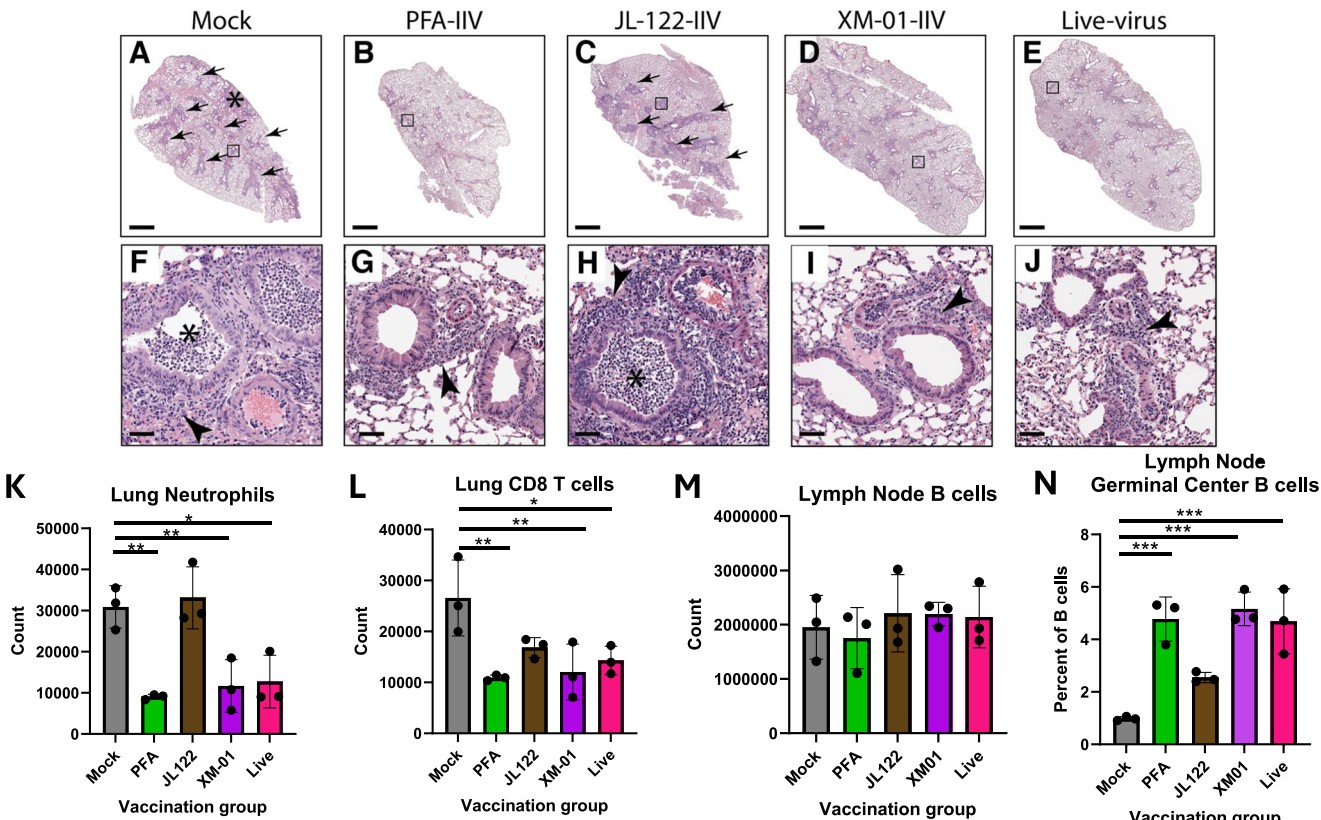

**Fig. 5 | Histological analysis and specific immune responses post vaccination and challenge. A–J** Representative H&E stained lung histology of vaccinated mice on day 5 post-challenge with live influenza virus. Scale bars: 1 mm (**A–E**) and 50 μm (**F–J**). **K–N** Immune cells detected by flow cytometry. **K** Infiltrating lung neutrophils (CD11b + Ly6G + ); **l** CD8 + T cells (TCRb+CD8 + ); **M** Total lymph node B cells

(CD19 + ) were quantified; and **N** percent germinal center B cells (CD19 + CD23 + GL7 + ). Data are presented as mean values +/− SD. Statistical analyses were performed with one-ANOVA and Dunnett's multiple comparisons test, which compared each IIV type to the mock control vaccinated group (N = 3 female mice; *P < 0.05, **P < 0.01, ***P < 0.001).

We observed that XM-01 physically and chemically changes the lipid bilayer of viral membranes by intercalating into the lipid bilayer, increasing membrane order and rigidity deep within the membrane, and increasing the phase transition temperature, all consistent with making the negative curvature events required for membrane fusion energetically unfavorable. One notable feature of XM-01 treatment is that the PC:PE ratios are drastically altered from the optimal range of 0.9–2.0 PC:PE[44] to the observed PC:PE ratio in XM-01 treated IAV of 10.97, mainly due to reduction in PE lipid species. This may be due to a significant portion of PE species in mammalian membranes being plasmalogens, which alleviate oxidative stress via the free radical scavenging capability of a vinyl-ether bond and *sn-2* polyunsaturated fatty acid acyl chain[45]. Other global lipid changes included a global increase in lipid unsaturation and an increase in lysolipids, both of which can contribute to making the negative membrane curvature required during the membrane fusion process energetically unfavorable[54]. These chemical changes within the membrane may explain the observed changes in its physical properties. We validated that the presence of XM-01 leads to inhibition of IV fusion with host cell membranes by utilizing lipid-mixing assays. As with LJ001 and related compounds, XM compounds can exhibit effects against both cells and viruses, although at different concentrations, indicating no specificity in the type of membrane targeted. However, cells possess robust mechanisms to repair damaged membranes, while viruses lack such lipid repair systems[55]. This differential capacity likely explains why viral membranes are significantly more sensitive to XM-01 than host cell membranes. Importantly, we do not anticipate that XM or related compounds would be used as therapeutic agents unless they were

specifically made to target viral membranes. This is because the abundant host cell membranes would likely sequester the XM compounds. The ability to preserve native antigenic structures while inactivating viral infectivity positions XM compounds as an ideal inactivation strategy for producing safe and effective WIV vaccines.

As a proof of principle, we utilized the unique properties of XM-01 to generate a whole inactivated virus (WIV) influenza vaccine. WIV vaccines offer advantages over subunit vaccines by presenting multiple viral proteins and genetic material to the immune system in the stoichiometric context of the naturally immunogenic viral particle[56]. However, many viral inactivation strategies for the generation of WIVs, such as heat, UV light, irradiation, or traditional PFA or BPL methods commonly used for IIV generation, are known to often damage the viral glycoprotein antigens during the inactivation process, which in turn can lead to ineffective, or even deleterious (e.g., for RSV), immune responses to vaccination[8–11,13,15,19,53,57]. To compare inactivation strategies, we selected PFA as a 'traditional' method due to its frequent use in laboratory and mechanistic studies of viral inactivation, and because it allows for standardization using the same live-virus stocks. Although β-propiolactone (BPL) is the standard inactivating agent for the newer most licensed influenza vaccines, direct comparisons with BPL were beyond the scope of this study. Future work including BPL-inactivated vaccines as a benchmark will further strengthen the translational relevance of our study. In total, we assessed IAV vaccination outcomes for PFA, XM-01, JL-122, and non-inactivated virus. Importantly, XM-01 fully inactivated IAV while leaving NA enzymatic activity and HA function unaffected. Notably, XM-01-IIV vaccination elicited robust neutralizing antibody (NAb) responses against both HA and NA,

achieving levels comparable to live-virus vaccination and surpassing those induced by traditional PFA-IIV immunization (Fig. 4B, C). While both anti-HA and anti-NA NAbs contribute to protection, the increase in NA NAbs is particularly notable, as these antibodies are thought to provide broader protection against multiple influenza strains[49]. Indeed, previous studies showed that passive transfer of serum containing antibodies specific only to NA completely prevented death in lethally-challenged mice[49]. Further, traditional IIV human vaccination does not seem to yield high NA NAb responses, which may contribute to its poor performance[49]. Interestingly, female mice developed stronger anti-HA and anti-NA NAb responses following fewer XM-01-IIV vaccinations compared to other vaccine types, including live-virus vaccination (Fig. S12). This suggests that XM-01-derived vaccines may enable faster seroconversion after a single booster dose. Sex-associated differences in immune responses to both influenza infection and vaccination have been previously reported, with female mice generally exhibiting more robust immunity, which may explain the observed effect (Supplementary Fig. 12)[58]. In a PRNT, sera from all vaccine groups effectively reduced viral plaque formation against the homologous Ca09 strain. However, only the XM-01-IIV and JL-122-IIV groups demonstrated measurable plaque reduction against a heterologous virus (WSN), highlighting their enhanced broad neutralization potential (Fig. 4D, E).

Histological analyses of various tissues and flow cytometric analyses of immune cells from vaccinated mice at 5 dpc supported the findings of enhanced induction of immunity by the XM-01 derived vaccine as compared to some of the other groups. Peribronchiolar and perivascular inflammation (arrowhead) was present in all mice with more severe inflammation in mice vaccinated with PBS and JL-122-IIV (arrows) and reduced inflammation in mice vaccinated with PFA-IIV, XM-01-IIV, and live-virus. Regions of interstitial pneumonia (Fig. 5A, asterisk) were only detected in mice vaccinated with PBS (Fig. 5A–E). Degenerate neutrophils within bronchiolar lumina (asterisk) were present in all female mice vaccinated with PBS or JL-122-IIV and absent in mice vaccinated with PFA-IIV, XM-01-IIV, and live-virus (Fig. 5F–J). Notably, histological examination of lung sections did not reveal quantitative differences between live-virus-vaccinated, XM-01-IIV, and PFA-IIV (Fig. 5A–J). Similarly, flow cytometric analysis of lung immune cells, which highlighted reduced infiltration of neutrophils and CD8[+] T cells, indicative of reduced lung inflammation upon challenge following vaccination with live-virus, XM-01-IIV, or PFA-IIV. However, the significant enhancement of NAbs to both HA and NA of the XM-01-IIV compared to all other IIV-vaccine groups correlated with reduced morbidity and mortality (Fig. 4). We speculate that these encouraging results may be due to both the quantity and quality of neutralizing antibodies, in turn due to increased glycoprotein preservation in the XM-01-IIV group, a major indicator of WIV vaccine potential[15]. The skewing of IgA antibody isotype within the live and XM-01 vaccinated animals likely played an important role in their protection from severe disease. IgA antibodies are mainly found in mucosal interfaces, such as the nasal passages, and serve to neutralize or slow initial infection and/ or subsequent propagation of the virus. Animal studies have shown that passive transfer of even non-NAbs against IV challenge can be fully protective against mortality[59]. This may explain why PFA-IIV had lower NAb titers yet maintained similar histological and specific immune cell responses as compared to the live-vaccinated and XM-01-IIV groups. Ultimately, however, the PFA-IIV and JL-122-IIV groups did not perform as well as the live-vaccinated or the XM-01-IIV groups in overall morbidity (weight loss) and mortality, indicating that not only the overall strength of the generated immune response impacts the outcome, but the specificity to key neutralizing antibody epitopes is likely critical for successful vaccination results. A limitation of our study may be that vaccine doses were standardized by total protein content and PFU equivalents, as opposed to other potential quantifications, such as HA content or particle-to-PFU ratio.

In summary, our results demonstrate a proof-of-principle study showing that membrane inactivators can support vaccine development. The XM class of antivirals effectively inactivates enveloped viruses by physically and chemically altering their membranes, while preserving glycoproteins. Our findings indicate that these properties can be leveraged to enhance whole inactivated virus (WIV) vaccine strategies. Further mechanistic chemical studies, such as those that optimize the rate of persulfide radical formation, may lead to future derivatives that possess improved antiviral activity, perhaps further improving vaccine development. The results of this proof-of-principle study may thus help improve inactivated influenza vaccine development. Evaluation of XM-01 inactivation for other enveloped viruses will be an important future research direction.

## Methods

### Small-molecule compounds tested
XM-01 and other sulfur-containing compounds tested were used in prior studies as $H_2S$ donors. These compounds were synthesized using known protocols[60]. LJ001 and JL122 compounds were synthesized at the University of California, Los Angeles (UCLA) by Dr. Michael Jung's group[23,27]. All stock solutions for these compounds were prepared in 100% DMSO, stored at −20 °C, and used within 6 months of reconstitution.

### Cell culture
HEK293T (ATCC) and PK13 cells (ATCC) were cultured in Dulbecco's modified Eagle's medium with 10% fetal bovine serum (FBS) (Gibco, Life Technologies). Vero cells (ATCC) were cultured in minimal essential medium alpha with 10% FBS. Human lung epithelial cells (A549, ATCC) and Madin–Darby canine kidney epithelial cells (MDCK, ATCC) were grown in complete DMEM containing 10% FBS, 100 IU/mL penicillin, 100 μg/mL streptomycin (Gibco, Life Technologies). MA104 cells were obtained from ATCC and grown in Dulbecco's modified Eagle's medium with 10% FBS, 100 IU/mL penicillin, 100 μg/mL streptomycin.

### Pseudotyped NiV/VSV viral infection assays
pNiV virus particles were incubated for 30 min with or without the indicated amounts of the compound or the corresponding vehicle, DMSO control. Vero cells were then infected with 10-fold dilutions of pseudotyped virus particles in infection buffer (PBS + 1% FBS) and incubated for 2 h at 37 °C. After 2 h, growth medium was added. 18–24 h post-infection, cells were lysed and an infinite M1000 microplate reader (Tecan Ltd) was used to measure luciferase activity. Three or more independent experiments were performed; error bars represent ± SD. Unpaired t-tests were performed using Graphpad PRISM software, with Bonferroni corrections.

### Cytotoxicity assay
Vero cells were incubated with each compound in the absence of FBS for 30 min to 24 h, as indicated, at the specified concentrations. This was followed by incubation with a cell counting kit (CCK-8) (Dojindo Molecular Technologies, Japan) for 1–2 h, and absorbance was measured at 450 nm using an infinite M100 microplate reader (Tecan Ltd). The quantity of the formazan dye produced when WST-8 (Dojindo) is reduced by dehydrogenases is directly proportional to the number of living cells (i.e., cell viability).

### Transmission electron microscopy (TEM) imaging
5 μL of concentrated VSV-NiV were pipetted onto a 200-mesh Formvar-coated nickel grid and allowed to settle for 20 min at RT. Excess liquid was removed by wicking with filter paper before coating the deposited sample with 5 μL of 1% uranyl acetate (UA) (Polysciences, Inc). After 2.5 min, excess UA was wicked off using filter paper and dried overnight in a desiccator. TEM micrographs of the samples were recorded

under high vacuum with an electron beam strength set at 200 kV using the FEI Technai G2 20 Twin TEM (FEI Corp., Hillsboro, OR).

### RNA stability assay

Briefly, we in-vitro transcribed CHIK reporter virus containing a luciferase gene in the hypervariable domain of nsP3. 20 µg of the nsP3:luciferase construct was treated with decapping enzyme at 1, 3, 10, 30, and/or 100 µM XM-01 concentrations for 1 h on ice, and then electroporated in BHK cells. After 2 h, enough time to initiate translation of nsP3 and luciferase, cells were lysed and luciferase activity measured.

### Detection of protein conformations by flow virometry

Pseudotyped NiV (pNiV) virions were incubated for 30 min with XM-01 at 4 °C, then washed by ultracentrifugation with NTE buffer (150 mM NaCl, 40 mM Tris-HCl at pH 7.5, and 1 mM EDTA) at 110,000 x RCF for 2 h. The treated virus was resuspended in NTE buffer, then stained as previously described[31]. We used conformational anti-NiV F and/or anti-NiV G specific rabbit primary antibodies (Anti NiV F Ab 66, or anti-NiV G Ab 213)[29,32,33] at 1:100 dilution for 1 h, followed by a FACS buffer (1% FBS in PBS) wash and incubation with secondary Alexa 647 goat anti-rabbit antibodies (Life Technologies, NY) for 30 min followed by one more FACS buffer wash. We then measured the relative levels of antibody binding through flow virometry, using a Guava easyCyte8HT flow cytometer (EMD Millipore, MA)[31]. Background mean fluorescence intensity (MFI) was obtained by binding equal concentrations of primary and secondary reagents to mock pseudotyped VSV virus, then subtracted from the MFI of pseudotyped NiV/VSV virions.

### Effect of XM-01 on viral membrane physical properties by ESR spectroscopy

Lipids 1-palmitoyl-2-oleoyl-glycero-3-phosphocholine (POPC), 1-palmitoyl-2-oleoyl-glycero-3-phosphoglycerol (POPG), lipid acyl chain spin-labeled 16PC, and head group spin-labeled DPPTC, all purchased from Avanti Polar Lipids (Alabaster, AL), and cholesterol, purchased from Sigma (St. Louis, MO), were used without further modification.

### Electron spin resonance

Prepared MLVs (POPC, POPG, and 0.5% (mol: mol) spin-labeled lipids) were resuspended and hydrated in a buffer consisting of 5 mM HEPES, 10 mM MES, 150 mM NaCl (pH 7) for 2 h at RT. Varying amounts of XM-01 (1 mg/mL in DMSO) were added to the MLV dispersion for 1 h at RT, along with the vehicle control. ESR spectra of ultracentrifuged MLVs were collected on an ELEXSYS ESR spectrometer (Bruker Instruments, Billerica, MA) at X-band (9.5 GHz) using an N2 Temperature Controller (Bruker Instruments, Billerica, MA). The ESR spectra were analyzed by the NLLS fitting program based on the stochastic Liouville equation[39,41] using the MOMD or Microscopic Order Macroscopic Disorder model, as in previous studies[35–37,61–64]. Each experiment (and subsequent fit) was repeated 2 or 3 times to check reproducibility and estimate experimental uncertainty.

### XM-01 decomposition

Butyl amine (47.4 mg, 0.65 mmol) was added to a solution of XM-01 (50 mg, 0.13 mmol) in $CH_2Cl_2$ (10 mL). The reaction mixture was stirred for 2 h at RT, then concentrated and subjected to column chromatography (30% Ethyl acetate/Hexane) to separate the products (33 mg) as a mixture of disulfide and polysulfides.

### NMR analysis

*Disulfide:* $^1$H NMR (300 MHz, Chloroform-d) δ 7.88–7.75 (m, 4H), 7.59–7.49 (m, 2H), 7.47–7.34 (m, 4H), 7.11 (d, J = 7.3 Hz, 2H), 5.07 (dt, J = 7.3, 5.1 Hz, 2H), 3.78 (s, 6H), 3.35 (d, J = 5.1 Hz, 4H); HRMS (ESI) m/z calcd. for $C_{22}H_{25}N_2O_6S_2$ [M + H]$^+$ 477.1154, found 477.1148.

*Polysulfides:* $^1$H NMR (300 MHz, Chloroform-d) δ 7.95–7.70 (m, 4H), 7.60–7.33 (m, 6H), 7.24–7.02 (m, 2H), 5.19–4.95 (m, 2H), 3.89–3.69 (m, 6H), 3.65–3.42 (m, 2H); HRMS (ESI) m/z calcd. for trisulfide $C_{22}H_{25}N_2O_6S_3$ [M + H]$^+$ 509.0875, found 509.0868; tetrasulfide $C_{22}H_{25}N_2O_6S_4$ [M + H]$^+$ 541.0595, found 541.0587; pentasulfide $C_{22}H_{25}N_2O_6S_5$ [M + H]$^+$ 573.0326, found 573.0313; hexasulfide $C_{22}H_{25}N_2O_6S_6$ [M + H]$^+$ 605.0037, found 605.0053.

### Lipidomics studies

40 mL of ~1 × 10$^5$ PFU/mL A/California/04/2009 was treated with either 0.02% PFA, 1 mM XM-01, or 10% DMSO, for 4 h at RT under steady rocking using an Orbitron shaker. Samples were then cleared at 1200 RCF for 10 min at 4 °C, supernatant was taken, and virus particles were collected over a 20% sucrose gradient by ultracentrifugation at 110,000 x RCF for 90 min at 4 °C. Supernatant was removed, and the virus was resuspended in minimal NTE buffer before performing lipid extraction. Viral membrane lipids were extracted using a modified liquid-liquid extraction procedure as described previously[63]. Extracted lipids were analyzed by LC-MS/MS, as described previously[63].

### Influenza A virus-liposome membrane binding

Liposomes were prepared by combining 220 µl of 10 mM POPC, 160 µl of 10 mM cholesterol, and 20 µl of 10 mM total gangliosides (Avanti Polar Lipids) dissolved in chloroform-methanol (2:1), resulting in a ratio of 55 mol% POPC, 40 mol% cholesterol, and 5 mol% total gangliosides. The solvent was removed by evaporation, first using a stream of argon gas and, second, a vacuum overnight. The lipid film was resuspended in 500 µl of HNE buffer (10 mM HEPES, pH 7.4, 150 mM NaCl, 1 mM EDTA), warmed to 60 °C, vortexed, processed using 15 freeze-thaw cycles, and extruded 10-times through a 100-nm polycarbonate membrane (Whatman) at 60 °C using a mini-extruder set (Avanti Polar Lipids).

Sucrose-purified influenza A virus A/WSN/33 (1 × 10$^7$ PFU/ml) was diluted 10-times in HNE buffer and incubated either with 1 mM XM-01 and 10% (v/v) DMSO or with 10% (v/v) DMSO (control) at RT for 4 h. Virus-XM-01 and virus-DMSO were mixed with liposomes in a 3:2 volume ratio and incubated for 20 min at 37 °C. Addition of 10 nm Protein A Gold (Aurion) and plunge freezing into liquid ethane cooled to −183 °C. Plunge freezing was performed using Leica GP2 with chamber humidity of 80% and temperature 25 °C, using a blotting time of 3 s.

### Viral membrane dye leakage assay

IV aliquots were incubated for 4 h with XM-01 (at a final concentration of 1 mM) or DMSO (vehicle; at an equivalent volume) at RT. Post incubation, free XM-01 or DMSO was removed by centrifugation at 735 × g using Microspin™ G-25 columns (Cytiva, Millipore Sigma, St. Louis, MO). All virus aliquots were then diluted 1:5 (vol/vol) with HEPES buffered saline (HBS; 10 mM HEPES, 50 mM Na citrate, 150 mM NaCl, pH 7.5). A 10-µl aliquot of each of the diluted virus sample was incubated with a 20-µl aliquot of SRB dissolved in HBS at a concentration of 25 mM at room temperature in the dark for 24 h. This corresponds to a quenched or semiquenched state of SRB. After 24 h, the virus samples were diluted by the addition of 70 µl of HBS, mixed well, and free SRB was removed using two centrifugation steps using Microspin™ G-25 columns, as described above. The SRB-loaded virus samples produced this way were immediately used to check for any effect of XM-01 on virus membrane integrity. For this, fluorescence intensity at excitation and emission wavelengths of 565 and 586 nm, respectively, of the SRB-loaded virus samples was recorded as a function of time using a Varian Cary Eclipse fluorescence spectrophotometer (Agilent Technologies, Santa Clara, CA). The reaction mixture consisted of 995 µl of HBS prewarmed to a temperature of 37 °C, to which 5 µl of the SRB-loaded virus aliquot was added and mixed well using a magnetic stirrer. Excitation and emission slit widths of 2.5 and 5 nm, respectively, and a

PMT voltage of 850 V was used to record fluorescence intensity till -190 s. Then, 50 μl of 10% Triton X-100 was used to permeabilize the viral membranes, which triggered release and complete dequenching of the encapsulated SRB molecules. Fluorescence intensity values were normalized for each virus sample (untreated, XM-01 treated, and DMSO-treated) with respect to the intensity value at the start of the experiment (t = 0 s).

## Lipid-mixing based bulk fusion assays

Virus-host fusion was reconstituted in a cuvette by using IV and PMVs from HEK293T cells as the virus and host component, respectively. PMVs were prepared from confluent HEK293T cells plated in 10-cm plates by treatment with 25 mM formaldehyde (Sigma-Aldrich) and 2 mM dithiothreitol (Sigma-Aldrich) in 4 ml of PMV buffer (10 mM HEPES, 2 mM CaCl₂, 150 mM NaCl, pH 7.4), as described previously[64]. IV aliquots (untreated and treated with XM-01 or XM-01 incubated with a -1000-fold excess of the radical quencher α-tocopherol) were prepared as described above and diluted 1:5 (vol/vol) with HBS. A 50-μl aliquot of each of the diluted virus sample was sonicated with a 0.7-μL aliquot of lipophilic R18 fluorophore dissolved in ethanol at a concentration of 0.5 mg/mL at 20 °C in the dark for 25 min. Then, free R18 was removed by centrifugation using Microspin™ G-25 columns, as described above. The R18-labeled virus aliquots were confirmed to consist of enough fluorophore concentration, such that R18 was in a quenched condition. Fusion of R18-labeled virus samples and host membrane mimicking GPMVs was triggered by low pH conditions (pH <4.5) achieved with the addition of predetermined aliquots of 4 (M) HCl.

Fluorescence intensity at excitation and emission wavelengths of 560 and 590 nm, respectively, of the R18-labeled virus was recorded as a function of time using a Varian Cary Eclipse fluorescence spectrophotometer. Excitation and emission slit widths of 2.5 and 5 nm, respectively, and a PMT voltage of 850 V was used for fluorescence measurements. The reaction mixture consisted of 694 μl of PMV buffer pre-warmed to a temperature of 37 °C, to which 1 μl of R18-labeled virus aliquot was added and mixed well using a magnetic stirrer. Baseline fluorescence intensity measurements were collected till t = 90 s, followed by the addition of 5 μl of 4 (M) HCl that lead to the pH drop required for IV fusion. After the pH drop, fluorescence intensity values were recorded till the values became asymptotic, followed by addition of 10 μl of 10% Triton X-100 to completely permeabilize the viral membrane. Fluorescence intensity values were subtracted from the baseline fluorescence intensity at t = 0 s and then normalized for each virus sample with respect to the fluorescence intensity value after detergent-induced permeabilization. Therefore, the normalized fluorescence intensity before the permeabilization step provided a quantitative estimate of the extent of bulk fusion in each case.

## Estimation of XM-01 partitioning in viral membranes by absorption spectroscopy

IV aliquots were incubated for 4 h with XM-01 (at a final concentration of 1 mM) or DMSO (vehicle; at an equivalent volume) at room temperature. Post incubation, free XM-01 or DMSO was removed using Microspin™ G-25 columns, as described above. Absorption spectra of IV treated with XM-01 or DMSO, along with that of XM-01 dissolved in DMSO at concentrations of 0.05, 0.1, 0.25, 0.5, 0.75, 1, and 2 mM, were recorded using a plate reader. Spectra were recorded in three technical replicates, averaged, and background corrected by subtracting the absorption spectra of appropriate blank samples. Spectra were moderately smoothened using the adjacent-averaging program in Microcal Origin v8.0 (OriginLab, Northampton, MA) while ensuring negligible smoothening-induced changes in spectral shape. Area under the curve (AUC) was calculated for all spectra (except for XM-01 in DMSO at concentrations of 0.05 and 0.1 mM, which were indistinguishable from the y = 0 baseline and represented the instrumental detection limit). A calibration curve for AUC vs XM-01 concentration was generated and

subjected to a linear fit, which was then employed to quantify the amount of XM-01 partitioned into IV membranes.

## Cryo-electron tomography

Cryo-electron tomography was performed using a Krios cryo-TEM (Thermo Fisher Scientific) operated at 300 keV, equipped with a post-column Quantum Gatan Imaging energy filter (Gatan) and K3 direct electron detector (Gatan) with an energy slit set to 20 eV. Tilt series were acquired using a dose-symmetric tilting scheme[65] with a nominal tilt range of 60° to −60° with 3° increments using the PACE tomography scripts[66] in SerialEM[67]. Records were acquired with target focus −4 μm and 3e-/Å2 electron dose at a magnification of 33,000× (pixel spacing of 2671 Å). Beam-induced sample motion and drift were corrected using MotionCor2[68]. Tilt series was aligned using fiducials, and tomograms were reconstructed in batch using R-weighted back projection algorithm using dose-weighting filter and SIRT-like filter 7 in the IMOD software package[69].

## Rotavirus infection assays

18β-glycyrrhetinic acid (GRA) (Sigma-Aldrich) stock solutions were prepared to a concentration of 100 mg/mL in DMSO, and aliquots were stored at −80 °C. Stock solutions were diluted to working concentrations in DMEM without FBS. For the control compounds, MA104 (ATCC) cells were treated for 6 h with 25 μg/mL GRA. Viability was measured with the Promega CellTiterGlo Assay according to the manufacturer's protocol, with digitonin as the control for 100% cytotoxicity. Data shown are representative of two experiments, with each concentration tested in triplicate in each experiment. Error bars indicate SEM. To test XM-01, after MA104 cells were infected with 8.9 × 10⁵ pfu/well of trypsin-activated bovine rotavirus strain NCDV. Mock-infected wells received 50 μl of 0% M199 vehicle media. 50 μl of fresh 2X control and experimental compounds were added, and at 18 h post-infection, the cells were fixed for 10 min with 80% acetone[70].

## Plaque assay for measurement of RSV infections

Human respiratory syncytial virus (RSV A2 strain) was propagated on CV 1 cells (ATCC) and purified by centrifuging two times on discontinuous sucrose gradients as described previously[71]. XM-01 (10 or 30 μM as indicated) was incubated with purified RSV at RT for 45 min or 2 h before infecting A549 cells at multiplicity of infection (MOI) of 0.5 or 0.01. Briefly, the XM-01 pre-treated RSV was adsorbed onto the cells in serum-free, antibiotic-free OPTI-MEM medium (Gibco) for 1.5 h at 37 °C. Following adsorption, A549 cells were washed with PBS and the infection was continued for 16 h in the presence of XM-01 before collecting the supernatant. A plaque assay was performed to determine the viral titer (pfu/mL) in the collected supernatant. Briefly, CV-1 cells were infected with serial dilutions of the culture supernatant in a 12-well plate as described above. After 1.5 h, the cells were washed with PBS and medium was replaced with 1% methylcellulose in complete growth medium. Plaques were stained after 24–48 h with 1% crystal violet and counted to determine the viral titer.

## Plaque assay for measurement of HSV-1 infections

XM-01 was incubated for 30 min with HSV-1 KOS (100 PFU/well), and then the mix was added to Vero cells. At 3 h post-inoculation, the medium was removed. At 18–24 hpi, culture medium was removed, and cells were fixed with ice-cold methanol-acetone solution (2:1 ratio) for 20 min at 20 °C and air dried. Virus titers were determined by immunoperoxidase staining with anti-HSV polyclonal antibody HR50 (Fitzgerald Industries, Concord, MA)[72,73].

## Plaque assay for measurement of A/WSN/33 and Ca/04/09 H1N1 influenza virus infections

The A/WSN/33 strain of influenza virus was serially diluted and then treated with either DMSO, 30 μM XM-01, or 1 μM LJ001 for 30 min at

4 °C, Ca/04/09 H1N1 was inactivated for 4 h at RT. Treated dilutions of virus were titrated on MDCK cells by standard plaque assay and plaque forming units were stained with crystal violet and counted 3–5 days post-infection[48].

### Influenza virus (IV) propagation and IIV passage in egg culture

All infections were done in certified pathogen free eggs following standard protocols[47]. Briefly, eggs (10–12 days old) were surface decontaminated in a biosafety cabinet, then 100 µl of inactivated virus or a low dose (-1 × 10^3 PFU/mL) of A/California/04/2009 was injected into the amniotic fluid. The injection site was sealed with glue and placed in an egg incubator for 72 h. Eggs were moved to 4 °C for 24 h before collecting amniotic fluid. Amniotic fluid was cleared by centrifugation ($1000 \times g$ for 5 min) and supernatant was aliquoted and snap-frozen in liquid nitrogen before being stored at −80 °C. Treated IV was passaged as described above, and amniotic fluid was assayed for infectious virus or passaged again in eggs up to three times to propagate any infectious particles.

### Vaccine generation and vaccination

A/California/04/2009 (-10 µg total influenza protein per vaccine) was thawed on ice before being incubated for 4 h at RT with 1 mM of XM-01 in 10% DMSO, 10 µM JL-122 in 10% DMSO, 0.02% PFA, or a mock control with vehicle only (10% DMSO). The solutions were then mixed 1:1 with alum for 30 min under light rocking before 100 µL was injected intramuscularly to mice. Vaccine doses were prepared to contain approximately 10 µg total protein per mouse vaccination, as measured by a BCA assay.

### Hemagglutination inhibition (HI) assay

Serum samples in HI buffer were added to a V-bottom 96-well plate, and diluted by 2-fold serial dilutions, then 8 HA units of Ca/04/2009 H1N1 in HI buffer was added. The plate was incubated for 30–45 min at RT, followed by the addition of 0.8% rooster blood in HI buffer. Results were read 30–45 min later, and the HI titer value was read as the inverse of the lowest dilution of serum that completely inhibited hemagglutination[47]. Three or more independent experiments were performed; error bars represent ±SD. Unpaired t-tests were performed using Graphpad PRISM software, with Bonferroni corrections.

### Neuraminidase inhibition (NI) assay

NI assays were performed using a protocol adapted from Leang and Hurt[48]. Briefly, sera taken from mice was assayed using 2-fold serial dilutions, then incubated with A/California/04/2009 virus in flat-bottom plates in assay buffer for 30–45 min at RT before the addition of the 20-(4-methylumbelliferyl)-α-D-N-acetylneuraminic acid (MUNANA) substrate. The sealed plate was incubated at 37 °C for 1 h before the addition of the stop solution. Results were read using a Tecan Spark plate reader set to 355 nm excitation, measuring absorbance at 460 nm. Three or more independent experiments were performed, error bars represent ± SD. Unpaired t-tests were performed using Graphpad PRISM software, with Bonferroni corrections.

### Antibody isotypes

Serum samples were undiluted, diluted 1:10, or 1:100, then analyzed for their isotype with the ProcartaPlex Mouse Antibody Isotyping Panel 2 7-plex (ThermoFisher Scientific), and then read on the MagPix (Luminex), and analyzed and graphed in Graphpad Prism Software. Data were analyzed with Two-way ANOVA and Dunnett's multiple comparison test, which determined significance from each vaccination group compared to Mock. $*p < 0.05$, $***p < 0.001$, $****p < 0.0001$.

### Histology

For histologic examination, lung tissues were collected directly after euthanasia and placed in 10% formalin for >72 h. Following paraffin embedding, 4 µm tissues sections were prepared and stained with hematoxylin and eosin (H&E).

### Flow cytometry

Lymphocytes were extracted from lung, mediastinal lymph node, and spleen, as previously described[68]. Cells were stained in 30 µL of 1:200 antibodies in PBS for 20 min at RT, washed, and then resuspended for flow cytometry. Fluorescent antibodies used for the study are listed as target (clone; catalog #). Antibodies against mouse antigens included CD16/32 (93; 14-0161-85) Fc block, CD3 (145-2C11; 45-0031-82), CD45 (30-F11; 56-0451-83), CD8 (53-6.7; 11-0081-82), CD19 (eBio1D3(1D3); 11-0193-82), CD23 (B3B4; 25−0232-82), GL-7 (GL-7(GL7); 48-5902-82) from Thermo Fisher Scientific. Ly6G (1A8; 565964), CD11b (M1/70; 563015) from BD Biosciences. Additionally, fixable viability dye (65-0866-14) was used to differentiate live/dead cells. Cells were analyzed on the BD Biosciences FACSymphony and analyzed with FlowJo Software (BD). Gating strategies are outlined in Supplemental Fig. 13.

### Animal care

All protocols were performed under BSL-2 conditions and approved by the Institutional Animal Care and Use Committee at Cornell University (IACUC # 2017-0108). Intranasal virus administration was performed under anesthesia, and all efforts were made to minimize animal suffering. Eight-week-old C57BL/6 mice were used for this study. Mice were intranasally inoculated with $1 \times 10^3$ PFU/animal (5 LD$_{50}$). Following challenge, mice were monitored and weighed daily and euthanized at predetermined humane criteria following approved protocols, generally when weight loss reached 20% from day of challenge or mice became moribund with a clinical score >3 on a 5-point scale. For each experimental condition, groups of mice (n = 10 per group, five males and five females) were challenged and monitored in parallel within a single experiment. Data reflect results consistent with standard practices in the field and were representative for each condition, allowing for statistical comparisons.

### Reporting summary

Further information on research design is available in the Nature Portfolio Reporting Summary linked to this article.

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

## Acknowledgements

We thank the rest of the Aguilar and Xian laboratories, as well as the WSU and Cornell University virology groups for helpful discussions. We thank the Infectious Diseases Imaging Platform (IDIP) at the Center for Integrative Infectious Disease Research Heidelberg and the cryo-EM network at the Heidelberg University (HD-cryoNET) for support and assistance. The authors gratefully acknowledge the data storage service SDS@hd supported by the Ministry of Science, Research, and the Arts Baden-Württemberg (MWK), the German Research Foundation (DFG) through grant INST 35/1314-1 FUGG and INST 35/1503-1 FUGG. This project was supported by grants NIH AI109022 and NIH AI156731 to HAC, NIH HL116571 to MX, NIH AI083387 to SB, and NIH AI119159 to AVN. A.P. was supported by the supplemental diversity award attached to grant NIH HL116571 to MX. A.A. was supported by grants NIH AI120701 and AI138570. D.W.B. was supported by T32EB023860. E.M.C. and J.L.Z. were supported by the NIH training grant T32GM008336, and MCJ by the NIH training grant T32ODO011000. B.I. was supported by NIGMS R25GM125597. J.W.J. acknowledges the University of Maryland School of Pharmacy Faculty Start-up funds and University of Maryland School of Pharmacy Mass Spectrometry Center (SOP1841-IQB2014). Work of P.C. was supported by a research grant from the Chica and Heinz Schaller Foundation (Schaller Research Group Leader Programme) and the Deutsche Forschungsgemeinschaft (DFG, German Research Foundation) project no. 240245660–SFB1129.

## Author contributions

D.W.B., A.P., S.P., I.A.M., S.X., B.I., J.S., M.J., A.L.L., E.M.C., S.E., B.C., E.R., Q.L., Y.Y.Y., A.M., H.B., O.S., J.L.Z., N.K.S., S.J.-B., S.M.P., C.Y., Y.Z., Z.J.M., C.X., M.J.J., G.V.d.W., S.M., M.S., A.G.G., M.H., S.B., A.V.N., J.H.F., A.A., S.D., P.C., J.W.J., M.X. and H.C.A. designed and/or performed experiments. D.W.B., A.P., S.P., I.A.M., E.M.C., J.W.J., P.C. and H.A.C. wrote the manuscript and D.W.B., A.P., S.P., I.A.M., J.L.Z., S.D., M.J.J., G.V.d.W., S.M., M.S., A.G.G., A.A., J.W.J., M.H., S.B., A.V.N., J.H.F., P.C., M.X. and H.A.C. edited the manuscript.

## Competing interests

HAC, DWB, and IAM are inventors on patent applications related to XM-01 antivirals and XM-01–generated vaccines filed by Cornell University. The remaining authors declare no competing interests.

## Additional information

[1]Department of Microbiology and Immunology, College of Veterinary Medicine, Cornell University, Ithaca, NY, USA. [2]Department of Chemistry, College of Arts and Sciences, Washington State University, Pullman, WA, USA. [3]Robert Frederick Smith School of Chemical and Biomolecular Engineering, College of Engineering, Cornell University, Ithaca, NY, USA. [4]Department of Chemistry, Brown University, Providence, RI, USA. [5]Department of Chemistry and Chemical Biology, College of Arts and Sciences, Cornell University, Ithaca, NY, USA. [6]Department of Chemistry and Biochemistry, University of California, Los Angeles, CA, USA. [7]Paul G. Allen School for Global Animal Health, College of Veterinary Medicine, Washington State University, Pullman, WA, USA. [8]Department of Veterinary Microbiology and Pathology, College of Veterinary Medicine, Washington State University, Pullman, WA, USA. [9]Department of Biology, Indiana University, Bloomington, IN, USA. [10]Center for Vaccines and Immunity, The Research Institute at Nationwide Children's Hospital, Columbus, OH, USA. [11]School of Molecular Biosciences, College of Veterinary Medicine, Washington State University, Pullman, WA, USA. [12]College of Letters, Sciences and Professional Studies, Montana Technological University, Butte, MT, USA. [13]Department of Virology, Center for Infectious Diseases, Uni-Heidelberg University Hospital, Heidelberg, Germany. [14]Department of Pharmaceutical Sciences, School of Pharmacy, University of Maryland, Baltimore, MD, USA. [15]Department of Microbiology, Immunology, and Molecular Genetics, David Geffen School of Medicine and College of Life Sciences, University of California, Los Angeles, CA, USA. [16]These authors contributed equally: David W. Buchholz, Armando Pacheco. ✉e-mail: HAguilarCarreno@mednet.ucla.edu

