## [Peer Review file · Nature Communications]

Sulfur-Containing Class of Broad-Spectrum Antivirals Improves Influenza Virus Vaccine Development

Corresponding Author: Professor Hector Aguilar

Version 0:

Reviewer comments:

Reviewer #2

(Remarks to the Author)

The authors partially addressed the critiques from the previous round of review. As also pointed out by other reviewers, the main selling point of this study is the discovery of XM compounds as chemical inactivators of enveloped viruses. It was proposed that upon activation, XM-01 produces persulfide radical, which leads to cross-linking of viral lipids. The authors provided convincing experimental results from SPR and LCMS to support the mechanism of action. Since there are several such chemicals that are current in use or being developed, including paraformaldehyde, β -propiolactone, hydrogen peroxide, psoralens + UV light, binary ethyleneimine, ascorbic acid (vitamin C) with Cu^{2+} , epoxides and aziridines and others, the novelty and significance of XM-01 need to be justified. Mechanistically, XM-01 produces radicals leading for lipid modification. Several other compounds have been reported acting as a similar mechanism, such as LJ001 and JL122. Despite the superior in vitro activity of LJ001 and JL122, this series of compounds has never moved forward in the translational development pipeline.

Additional comments are:

1. For a chemical inactivator used to prepare viral vaccines, XM-01 needs to demonstrate superior safety. This is generally a concern for the current chemical inactivators such as paraformaldehyde and β -propiolactone. Have the authors tested 1) the minimal amount of XM-01 needed for complete inactivation? 2) the leftover amount of XM-01 in the virus sample 3) the in vivo toxicity of XM-01.
2. As pointed by reviewer 4, were the viruses completely inactivated by XM-01 treatment? Have the authors performed amplification assay for the inactivated viruses to see no active viruses can be recovered?
3. The broader application of XM-01 as an inactivator to produce vaccines for other viruses has not been demonstrated. The authors might consider tuning down the claim.

Reviewer #3

(Remarks to the Author)

The paper by Pacheco et al describes the characterization of a new antiviral, XM01, and its use for the production of a whole inactivated influenza virus (WIV) vaccine. The authors describe convincingly that the drug is capable of inhibiting various membrane viruses, and provide data indicating that it most probably does so by disturbing the lipid membrane without damaging the viral surface proteins. CM01-inactivated influenza virus (adjuvanted with alum) induced NA- and HA-inhibiting antibodies with neutralizing activity to homologous A/Cal and heterologous A/WSN/33 virus and protected mice from weight loss and death after homologous challenge. In these aspects, the CM01-produced vaccine was somewhat more potent than WIV produced by traditional formaldehyde treatment.

In a review of an earlier version of the paper, I raised several points of criticism and suggested several additional experiments. The authors did a remarkable effort to implement my feedback and addressed the earlier raised points to my satisfaction.

A minor point left: The authors used many different techniques in their experiments but the description of some figures is rather short and for readers less familiar with the technique used sometimes difficult to understand and interpret. For me, this is the case for the physico-chemical characterization of XM01.

Reviewer #4

(Remarks to the Author)

Manuscript title: New Class of Broad-Spectrum Antivirals Improves Influenza Virus Vaccine Development".

The manuscript focuses entirely on influenza viruses (IAV and IBV), particularly how the XM-01 to XM-05 compounds modulate viral infectivity and enhance inactivated influenza vaccine efficacy. The paper claims that XM-01 to XM-05 affect viral membranes by altering lipid composition, increasing membrane order, and inhibiting fusion. These mechanisms are supported by referenced techniques (e.g. GC-MS lipidomics, GP measurements) and align with current understanding of membrane-targeting antivirals.

The XM-01–inactivated IAV vaccines elicited higher IgA and IgG2a responses and neutralizing titers. Furthermore, mechanistic claims are well-supported with experimental data and relevant literature.

Major issues:

1. "Leaving viral glycoproteins and genomes largely unaffected" A strong mechanistic claim. it's important to say how this was assessed (e.g., Western blot, RT-qPCR).
2. Calling compound XM-01 "broad-spectrum" might be overstated. It should be clarified.
3. In "Lipidomics Results Interpretation" The paper reports major phospholipid class shifts. Please clarify if these changes reflect selective partitioning into viral envelopes or result from oxidation, metabolic disruption, or artifact of inactivation.
4. Comparison with PFA May not Be correct because PFA is not commonly used for licensed influenza vaccine production. Please explain. A stronger comparison would include a benchmark like BPL-inactivated vaccine for translational relevance.
5. In Mouse Model Data section—The challenge data (body weight, survival) are convincing, but it's unclear if multiple independent repeats were done. Clarify in methods
6. The manuscript uses PFU for viral challenge but does not determine the MLD₅₀, which is the standard way to express lethal challenge doses in influenza models.
7. The vaccine dose is expressed as PFU, but this doesn't reflect total antigen or HA content. PFU-to-particle ratio can vary. in rebuttal: Total protein was used (~10 µg per dose), but HA content wasn't directly quantified

REVIEWERS' COMMENTS

Reviewer #1 (No additional comments)

Reviewer #2 (Remarks to the Author):

The authors partially addressed the critiques from the previous round of review. As also pointed out by other reviewers, the main selling point of this study is the discovery of XM compounds as chemical inactivators of enveloped viruses. It was proposed that upon activation, XM-01 produces persulfide radical, which leads to cross-linking of viral lipids. The authors provided convincing experimental results from SPR and LCMS to support the mechanism of action. Since there are several such chemicals that are current in use or being developed, including paraformaldehyde, β -propiolactone, hydrogen peroxide, psoralens + UV light, binary ethyleneimine, ascorbic acid (vitamin C) with Cu^{2+} , epoxides and aziridines and others, the novelty and significance of XM-01 need to be justified. Mechanistically, XM-01 produces radicals leading for lipid modification. Several other compounds have been reported acting as a similar mechanism, such as LJ001 and JL122. Despite the superior in vitro activity of LJ001 and JL122, this series of compounds has never moved forward in the translational development pipeline.

Author response: We thank the reviewer for the overall positive review of our study. The reviewer is correct in that other chemicals are currently in use or being developed to inactivate viruses, and in fact we used one of them, paraformaldehyde, as a control to study whether the persulfide radical producing compound XM-01 was able to show superior properties for vaccine development over paraformaldehyde. Our results confirmed the utility of XM-01 for vaccine development. The reviewer is also correct in that other chemicals have been found to produce radicals, such as LJ001 and JL122. The reviewer is also correct is that these compounds have not moved forward in the translational development pipeline. However, LJ001 and JL122 have not been developed for vaccine development, only for therapeutic development. The novelty in our study is that the persulfide compounds such as XM-01 show ideal properties for vaccine development, and such properties are superior to those of LJ001 or JL122, which we used as control compounds.

Additional comments are:

1. For a chemical inactivator used to prepare viral vaccines, XM-01 needs to demonstrate superior safety. This is generally a concern for the current chemical inactivators such as paraformaldehyde and β -propiolactone. Have the authors tested 1) the minimal amount of XM-01 needed for complete inactivation? 2) the leftover amount of XM-01 in the virus sample 3) the in vivo toxicity of XM-01.

Author response: We thank the reviewer for these important points regarding the safety and use profile of XM-01 as a viral inactivator. Below, we address each aspect in turn:

1) Minimal amount of XM-01 for complete inactivation:

We optimized the viral inactivation protocol by testing various concentrations and exposure times. We found that treatment of influenza virus with 1 mM XM-01 for 4 hours at room temperature reliably resulted in complete inactivation, as confirmed by plaque assay and by three serial passages in embryonated chicken eggs with no detectable residual infectivity (see Fig. S10). Lower

concentrations were also evaluated during optimization, but 1 mM was chosen as the minimum concentration that ensured COMPLETE inactivation without compromising viral antigenic structure.

2) Residual XM-01 in virus samples post-treatment:

We quantified the amount of XM-01 remaining in the virus samples after inactivation by absorption spectroscopy. After incubation with 1 mM XM-01 for 4 hours, the amount of XM-01 associated with the viral membrane was measured to be approximately 0.31 mM (see Fig. S9). This residual, membrane-associated fraction represents the XM-01 present in the preparations used for animal studies, as no additional removal steps were performed. Importantly, experiments performed with the inclusion of this residual concentration or higher concentrations resulted in no adverse effects in the animal.

3) In vivo toxicity of XM-01:

To evaluate *in vivo* safety, we performed intramuscular injections of 1 mM XM-01 (higher than the residual XM-01 amount post virus inactivation) in mice and observed no weight loss or overt toxicity (see Fig. S11). Furthermore, mice vaccinated with XM-01-inactivated virus exhibited no adverse effects throughout vaccination and challenge experiments, supporting a favorable *in vivo* safety profile under the conditions used.

2. As pointed by reviewer 4, were the viruses completely inactivated by XM-01 treatment? Have the authors performed amplification assay for the inactivated viruses to see no active viruses can be recovered?

Author response: Absolutely! Regarding this important point, as described in the Results and Methods sections (stated in our resubmitted manuscript), we rigorously confirmed complete inactivation of influenza virus by XM-01 treatment. Specifically, following treatment of virus with 1 mM XM-01 for 4 hours at room temperature, samples were tested for residual infectivity by standard plaque assay and, importantly, by three consecutive amplification passages in embryonated chicken eggs. In all cases, no viable virus was detected, indicating complete inactivation under these conditions (see Fig. S10). These results demonstrate that XM-01 treatment fully inactivates the virus, and that our preparations are free of recoverable infectious virus.

3. The broader application of XM-01 as an inactivator to produce vaccines for other viruses has not been demonstrated. The authors might consider tuning down the claim.

Author response: We appreciate the reviewer's suggestion and agree that our current experimental data are limited to influenza virus. While our antiviral screening indicates XM-01 has broad inhibitory activity against multiple enveloped viruses, we have not yet directly demonstrated its use as an inactivator for vaccine production beyond influenza virus. We have revised the manuscript text to clarify that our data specifically support the use of XM-01 for influenza virus inactivation, and we now refer to the potential broader application as a future direction, rather than a demonstrated outcome.

Changes in text:

Lines 304-306: Changed from "We report a new class of broad-spectrum antivirals with broad inhibitory properties against enveloped viruses." To "We report a new class of broad-spectrum antivirals with

broad inhibitory properties against *multiple* enveloped viruses *in cellulo*, and demonstrate its use as a viral inactivator for influenza vaccine development.”

Lines 401-403: Changed from “The results of this proof-of-principle study may thus help us better prepare against current endemic as well as emerging enveloped viral pathogens and their potential pandemics, which pose a major threat to human and animal global health.” to “The results of this proof-of-principle study may thus help improve inactivated influenza vaccine development. Evaluation of XM-01 inactivation for other enveloped viruses will be an important future research direction.”

Reviewer #3 (Remarks to the Author):

The paper by Pacheco et al describes the characterization of a new antiviral, XM01, and its use for the production of a whole inactivated influenza virus (WIV) vaccine. The authors describe convincingly that the drug is capable of inhibiting various membrane viruses, and provide data indicating that it most probably does so by disturbing the lipid membrane without damaging the viral surface proteins. CM01-inactivated influenza virus (adjuvanted with alum) induced NA- and HA-inhibiting antibodies with neutralizing activity to homologous A/Cal and heterologous A/WSN/33 virus and protected mice from weight loss and death after homologous challenge. In these aspects, the CM01-produced vaccine was somewhat more potent than WIV produced by traditional formaldehyde treatment.

In a review of an earlier version of the paper, I raised several points of criticism and suggested several additional experiments. The authors did a remarkable effort to implement my feedback and addressed the earlier raised points to my satisfaction.

A minor point left: The authors used many different techniques in their experiments but the description of some figures is rather short and for readers less familiar with the technique used sometimes difficult to understand and interpret. For me, this is the case for the physico-chemical characterization of XM01.

Author response: We thank the reviewer for this valuable feedback and for stating that “The authors did a remarkable effort to implement my feedback and addressed the earlier raised points to my satisfaction.” We agree that some of the descriptions related to the physico-chemical characterization of XM-01 could be enhanced for clarity, especially for readers less familiar with these methods. In response, we have revised the text to provide additional explanation of the experimental approaches, key principles, and interpretation of the results for the relevant physico-chemical assays (e.g., ESR spectroscopy, lipid-mixing assay, absorption spectroscopy). We hope these changes will make the figures and data more accessible to a broader audience. Examples of our changes to address this point include:

Lines 188-196: Changed from “Electron spin resonance (ESR) spectroscopy was used to evaluate large multilamellar vesicles (MLVs) treated with XM-01 using lipids with spin labels at various depths within the membrane. ESR spectroscopy provides a measure of the order in the microenvironment around a spin label (a chemical group that acts as a sensitive spatially localized reporter). ESR spectroscopy revealed that XM-01 intercalates deep into lipid bilayers, particularly at the hydrophobic membrane interior, as indicated by spectral shifts in spin-labeled lipids such as 1-palmitoyl-2-(16-doxyl stearoyl) phosphatidylcholine (16PC)” to “To investigate how XM-01 affects the physical state of viral membranes, we used electron spin resonance (ESR) spectroscopy. This technique enables the detection of changes in membrane order and fluidity by measuring the behavior of spin-labeled lipids incorporated at different positions within the bilayer. ESR analysis allows us to determine whether XM-01 increases membrane rigidity or alters membrane dynamics, both of which could impact viral fusion. Our results show that XM-01 intercalates deep into the lipid bilayer, particularly at the hydrophobic interior, as evidenced by shifts in the spectra of spin-labeled 16PC lipids. These findings indicate that XM-01 increases the order and rigidity of the viral membrane, which is likely to reduce membrane fusion capacity (Fig. 3A–B).”

Lines 198-202: Changed from “To quantify membrane structural changes, lipid order parameters (ΔS_0) were calculated for XM-01-treated bilayers at various molar ratios. Significant increases in ΔS_0 were

observed for 16PC, indicating enhanced order in the membrane interior, while no changes were detected for dipalmitoylphosphatidyltempocholine (DPPTC), which probes the lipid headgroup region (Fig. 3B).” to “To quantitatively assess these changes, we calculated the lipid order parameter (ΔS_0), a value that reflects the degree of organization and rigidity within the membrane. Higher ΔS_0 values correspond to greater membrane order. We observed that XM-01 treatment led to significant increases in ΔS_0 for 16PC-labeled bilayers, confirming that XM-01 promotes a more ordered and less fusogenic membrane state (Fig. 3B).”

Lines 219-223: Changed from “We quantified the amount of XM-01 that partitions into the viral membrane post-incubation by absorption spectroscopy. XM-01 concentration in IV membranes post-incubation with 1 mM XM-01 for 4 h at room temperature (RT) was calculated to be ~ 0.31 mM (indicating a buffer-to-membrane partition coefficient of ~ 0.3) by plugging in values of the area under the corrected absorption spectrum for XM-01 in IV membranes (Fig. S9, left) to the calibration plot (Fig. S9, right).” to “To determine how much XM-01 remains associated with the viral membrane after treatment, we utilized absorption spectroscopy. This technique measures the absorbance of light at specific wavelengths corresponding to XM-01 and allows us to estimate its concentration within the viral sample. After treatment with 1 mM XM-01, we calculated that approximately 0.31 mM XM-01 remains associated with the viral membranes (Fig. S9), reflecting the membrane-bound fraction that persists after incubation.”

Reviewer #4 (Remarks to the Author):

Manuscript title: New Class of Broad-Spectrum Antivirals Improves Influenza Virus Vaccine Development".

The manuscript focuses entirely on influenza viruses (IAV and IBV), particularly how the XM-01 to XM-05 compounds modulate viral infectivity and enhance inactivated influenza vaccine efficacy. The paper claims that XM-01 to XM-05 affect viral membranes by altering lipid composition, increasing membrane order, and inhibiting fusion. These mechanisms are supported by referenced techniques (e.g. GC-MS lipidomics, GP measurements) and align with current understanding of membrane-targeting antivirals.

The XM-01–inactivated IAV vaccines elicited higher IgA and IgG2a responses and neutralizing titers. Furthermore, mechanistic claims are well-supported with experimental data and relevant literature.

Author response: We thank the reviewer for the overall positive view of our manuscript, and particularly for stating that “mechanistic claims are well-supported with experimental data and relevant literature.”

Major issues:

1. “Leaving viral glycoproteins and genomes largely unaffected” A strong mechanistic claim. it’s important to say how this was assessed (e.g., Western blot, RT-qPCR).

Author response: We thank the reviewer for his point. The following section in text explains how these claims were assessed.

Lines 140-178: XM-01 perturbs viral membranes without affecting viral glycoproteins or RNA. To determine if viral infection is inhibited by XM-01 at viral entry or post-viral entry steps, we incubated Vero cells with XM-01 at different time points post- viral infection. Cells were infected with pNiV for 2, 4, 6, 8, 18, or 24 h, after which unbound pNiV was washed away from the cells, and 100 μ M XM-01 in media was added for the remainder of 24 h at 37°C. pNiV infection was then measured via luciferase activity (Fig. 2A). No inhibition by XM-01 was observed compared to the DMSO vehicle control at any time point, indicating that once the virus has entered cells, XM-01 does not exert inhibitory activity, consistent with XM-01 specifically inhibiting the viral entry step. We next incubated either Vero cells or pNiV for 30 min with XM-01 at various concentrations prior to viral infection (Fig. 2B). Viral infection assessed 24 hours post infection (hpi) was decreased only when pNiV virions, but not when cells, was pre-incubated with XM-01, suggesting a direct effect of XM-01 on virions rather than on host cells.

Further, to narrow down whether viral membranes or viral glycoproteins were affected by XM-01, we tested whether XM-01 affected the conformations of NiV-F or -G. pNiV virions were incubated with 10 μ M XM-01 for 30 min, followed by measurement of protein function or by binding of established conformational antibodies to NiV F or G via flow virometry, to determine potential conformational changes in F or G [29, 31-33]. As a control we used LJ001, a compound known to affect viral membranes but not viral glycoprotein conformations [23, 27]. In both cases, no statistically significant changes were observed in conformational antibody binding to either glycoprotein (Fig. 2C), indicating that native glycoprotein conformations remained conserved upon XM-01 treatment. To further test for potential effects on protein function following inactivation, we used influenza virus (IV) Ca09. When IV was treated with up to 1,000 μ M XM-01 for 4 h, neither neuraminidase (NA) nor hemagglutinin (HA) activity was significantly impacted as compared to the untreated control (Fig. 2D-E). These were the same conditions

confirmed to yield fully-inactivated influenza virus (Fig. 2D), and used for vaccine development as further described below for Figs. 4, 5, and S6.

To visually look for gross viral structural changes, we treated pNiV with 10 μ M XM-01 or LJ001 control compound for 30 min, then imaged particles via electron microscopy. Most viral particles treated with XM-01 or the LJ001 control displayed compromised membranes, altered morphologies, and frequent RNA spillage compared to the DMSO control, consistent with XM-01 and LJ001 affecting viral membranes (Figs. 2F, S4) [34, 35]. To test the effect of XM-01 on viral RNA integrity, we in vitro transcribed CHIK reporter virus, which contains luciferase in the hypervariable domain of non-structural protein 3 (nsP3). RNA was treated with different compounds including XM-01 and decapping enzyme, and then electroporated in BHK cells. After 2 h, enough time for the CHIK reporter to initiate translation of non-structural protein and the luciferase to be produced, cells were lysed and luciferase activity measured. No effects of XM-01 were observed for XM-01 on viral RNA (Fig. 2G). We also treated IV with 1 mM XM-01 for 4 h then measured viral binding to POPC/Chol/GM1 liposomes by cryoelectron microscopy. We observed that viral particles remain intact and there were no changes in virus binding to liposomes, indicating that XM-01 treatment maintains structure of the particle while preserving protein function (Fig. S5). Combined, these data indicate that XM-01 modulates viral entry by affecting viral membranes and not the viral glycoproteins or RNA.

2. Calling compound XM-01 "broad-spectrum" might be overstated. It should be clarified.

Author response: While XM-01 showed inhibitory activity against several different enveloped viruses in our *in cellulo* assays, we agree that the term "broad-spectrum" may be interpreted too generally. In response, we have clarified in the manuscript that XM-01 demonstrated activity against *multiple* tested enveloped viruses *in cellulo*.

Changes in text:

Lines 304-306: Changed from "We report a new class of broad-spectrum antivirals with broad inhibitory properties against enveloped viruses." To "We report a new class of broad-spectrum antivirals with broad inhibitory properties against multiple enveloped viruses *in cellulo*, and demonstrate its use as a viral inactivator for influenza vaccine development."

3. In "Lipidomics Results Interpretation" The paper reports major phospholipid class shifts. Please clarify if these changes reflect selective partitioning into viral envelopes or result from oxidation, metabolic disruption, or artifact of inactivation.

Author response: We thank the reviewer for this thoughtful question. The observed changes in phospholipid classes following XM-01 treatment most likely reflect chemical modification and selective oxidation of membrane lipids as a direct result of XM-01's radical-generating activity, rather than selective partitioning or metabolic disruption (since the experiments were performed on purified viral particles outside of cellular context). These results are consistent with the known chemical properties of persulfide-radical donors, which can oxidize susceptible lipids. We have clarified this interpretation in the revised Results section to avoid any potential confusion.

Lines 232-235: Added "These phospholipid shifts are most likely due to oxidative modification of membrane lipids by XM-01's radical-generating activity during inactivation, rather than selective

partitioning or metabolic effects, since all experiments were performed on purified viral particles in the absence of cellular metabolism.”

4. Comparison with PFA May not Be correct because PFA is not commonly used for licensed influenza vaccine production. Please explain. A stronger comparison would include a benchmark like BPL-inactivated vaccine for translational relevance.

Author response: We thank the reviewer for this important comment. We agree that β -propiolactone (BPL) is more commonly used for inactivated influenza vaccines in clinical settings, whereas paraformaldehyde (PFA) is primarily used in laboratory studies. In our study, we chose PFA as a comparator due to its frequent use in published research on viral inactivation and its well-characterized effects on antigenic structure, which provided a relevant benchmark for our mechanistic comparisons. We acknowledge that a great idea for a future direction would be a direct comparison with BPL, but such animal experiments would basically mean repeating all our animal studies, which took years to complete, and our current experimental and funding resources make this effort currently impossible. We have clarified this rationale in the revised manuscript and now explicitly state this limitation in the Discussion.

Changes in Text:

Lines 343-348: Changed “To compare inactivation strategies, we selected PFA as a ‘traditional’ method, as it closely resembles current IAV vaccination approaches and allows for standardization using the same live-virus stocks.” to “To compare inactivation strategies, we selected PFA as a 'traditional' method due to its frequent use in laboratory and mechanistic studies of viral inactivation, and because it allows for standardization using the same live-virus stocks. Although β -propiolactone (BPL) is the standard inactivating agent for the newer most licensed influenza vaccines, direct comparisons with BPL were beyond the scope of this study. Future work including BPL-inactivated vaccines as a benchmark will further strengthen the translational relevance of our study.”

5. In Mouse Model Data section—The challenge data (body weight, survival) are convincing, but it’s unclear if multiple independent repeats were done. Clarify in methods

Author response: We thank the reviewer for their careful reading. All mouse challenge experiments were performed using groups of 10 mice (5 males + 5 females), assayed in parallel within a single experiment, which is a standard approach for these studies. Each group included multiple animals, and the results were representative for each condition and allowed statistical comparisons. We have clarified this detail in the Methods section.

Changes to text:

Lines 661-664: Added “For each experimental condition, groups of mice (n=10 per group, five males and five females) were challenged and monitored in parallel within a single experiment. Data reflect results consistent with standard practices in the field and were representative for each condition, allowing for statistical comparisons.”

6. The manuscript uses PFU for viral challenge but does not determine the MLD_{50} , which is the standard way to express lethal challenge doses in influenza models.

Author response: We agree that expressing the challenge dose in terms of LD₅₀ is standard in influenza mouse models. As noted in the Results and Methods sections, we determined the LD₅₀ for our influenza virus stock and used a challenge dose of 5 LD₅₀ (corresponding to 1,000 PFU/mouse). We have now clarified this in both the Results and Methods sections to ensure it is clearly stated for readers.

Changes to text:

Line 658-659: Mice were intranasally inoculated with 1x10³ PFU/animal (5 LD₅₀).

7. The vaccine dose is expressed as PFU, but this doesn't reflect total antigen or HA content. PFU-to-particle ratio can vary. in rebuttal: Total protein was used (~10 µg per dose), but HA content wasn't directly quantified

Author response: We thank the reviewer for this important point. While we expressed the vaccine dose in terms of PFU equivalents (based on the infectious titer of the starting virus), the actual amount of antigen administered was standardized by total protein content, with each dose containing approximately 10 µg of protein. We did not directly quantify HA content or the PFU-to-particle ratio for these preparations, which we now acknowledge as a limitation. We have clarified this point in the revised Methods and Discussion sections.

Changes to text:

Lines 393-394: Added "A limitation of our study may be that vaccine doses were standardized by total protein content and PFU equivalents, as opposed to other potential quantifications, such as HA content or particle-to-PFU ratio."

Lines 615-616: Added "Vaccine doses were prepared to contain approximately 10 µg total protein per mouse vaccination, as measured by a BCA assay."